# MEIS-mediated suppression of human prostate cancer growth and metastasis through HOXB13-dependent regulation of proteoglycans

Calvin VanOpstall[1], Srikanth Perike[2], Hannah Brechka[1], Marc Gillard[3], Sophia Lamperis[2], Baizhen Zhu[3], Ryan Brown[2], Raj Bhanvadia[4], Donald J Vander Griend[2]*

[1]The Committee on Cancer Biology, The University of Chicago, Chicago, United States; [2]Department of Pathology, The University of Illinois at Chicago, Chicago, United States; [3]Department of Surgery, Section of Urology, The University of Chicago, Chicago, United States; [4]Department of Urology, UT Southwestern, Dallas, United States

**Abstract** The molecular roles of HOX transcriptional activity in human prostate epithelial cells remain unclear, impeding the implementation of new treatment strategies for cancer prevention and therapy. MEIS proteins are transcription factors that bind and direct HOX protein activity. MEIS proteins are putative tumor suppressors that are frequently silenced in aggressive forms of prostate cancer. Here we show that MEIS1 expression is sufficient to decrease proliferation and metastasis of prostate cancer cells in vitro and in vivo murine xenograft models. HOXB13 deletion demonstrates that the tumor-suppressive activity of MEIS1 is dependent on HOXB13. Integration of ChIP-seq and RNA-seq data revealed direct and HOXB13-dependent regulation of proteoglycans including decorin (DCN) as a mechanism of MEIS1-driven tumor suppression. These results define and underscore the importance of MEIS1-HOXB13 transcriptional regulation in suppressing prostate cancer progression and provide a mechanistic framework for the investigation of HOXB13 mutants and oncogenic cofactors when MEIS1/2 are silenced.

*For correspondence:
dvanderg@uic.edu

Competing interests: The authors declare that no competing interests exist.

## Introduction

Prostate cancer (PrCa) is the fifth leading cause of cancer-related death in men worldwide and is responsible for the highest incidence of male cancer in the United States (*Ferlay et al., 2015*; *Siegel et al., 2019*). While PrCa can progress slowly and remain relatively asymptomatic for years, some patients present with aggressive metastatic PrCa and a poor prognosis (*Barlow and Shen, 2013*; *Lin et al., 2009*). Further, it can be difficult to distinguish which men harbor indolent or aggressive tumors (*Culig, 2014*), particularly in patients with intermediate Gleason scores (*Gearman et al., 2018*). These features of PrCa pose a significant clinical problem.

One novel pathway to understand tumor etiology and disease progression as well as develop new treatments is through *HOXB13*, which exhibits germline mutation in a subset of familial PrCa. *HOXB13* is the predominant HOX factor that drives development and differentiation of prostate epithelial cells (*Brechka et al., 2017*). Germline mutations of *HOXB13* confer a substantial risk of PrCa, but mutation frequency is rare within the general population (*Brechka et al., 2017*). On the other hand, our prior studies show that prostate tumors frequently harbor downregulation of the transcription factors and HOX binding partners MEIS1 and MEIS2 (myeloid ecotropic viral integration site 1/2) (*Bhanvadia et al., 2018*; *Chen et al., 2012*). MEIS proteins function as critical transcriptional co-

**eLife digest** Decisions regarding the treatment of patients with early-stage prostate cancer are often based on the risk that the cancer could grow and spread quickly. However, it is not always straightforward to predict how the cancer will behave. Studies from 2017 and 2018 found that samples of less aggressive prostate cancer have higher levels of a group of proteins called MEIS proteins. MEIS proteins help control the production of numerous other proteins, which could affect the behavior of prostate cancer cells in many ways. VanOpstall et al. – including some of the researchers that performed the 2017 and 2018 studies – have investigated how MEIS proteins affect prostate cancer.

When prostate cancer cells are implanted into mice, they result in tumors. VanOpstall et al. found that tumors that produced MEIS proteins grew more slowly. Next, MEIS proteins were extracted from the prostate cancer cells and were found to interact with another protein called HOXB13, which regulates the activity of numerous genes. When the cells were genetically modified to prevent HOXB13 being produced, the protective effect of MEIS proteins was lost.

MEIS proteins work with HOXB13 to regulate the production of several other proteins, in particular a protein called Decorin that can suppress tumors. When MEIS proteins and HOXB13 are present, the cell produces more Decorin and the tumors grow more slowly and are less likely to spread. VanOpstall et al. found that blocking Decorin production rendered MEIS proteins less able to slow the spread of prostate cancer. These results suggest that MEIS proteins and HOXB13 are needed to stop tumors from growing and spreading, and some of this ability is by prompting production of Decorin.

This study explains how MEIS proteins can reduce prostate cancer growth, providing greater confidence in using them to determine whether aggressive treatment is needed. A greater understanding of this pathway for tumor suppression could also provide an opportunity for developing anti-cancer drugs.

factors during development and within adult tissues to bind HOX proteins and specify *HOX* gene targeting (*Merabet and Mann, 2016*). Most PrCa *HOXB13* mutations (including the original G84E mutation) are located within the MEIS-interacting domain, emphasizing the importance of MEIS/HOX interactions in prostate tumor biology.

We originally demonstrated that increased mRNA expression of *MEIS1* and *MEIS2* in PrCa is correlated with significantly longer overall survival in a large cohort of watchful waiting patients with mid-range Gleason scores (*Chen et al., 2012*). More recently, we and others demonstrated that patients harboring MEIS-positive tumors have a significantly favorable outcome; there is a step-wise decrease in both *MEIS1* and *MEIS2* expression as tumors progress to metastatic (*Bhanvadia et al., 2018*; *Jeong et al., 2017*; *Nørgaard et al., 2019*). These correlative findings provide support to a tumor-suppressive role for MEIS1 and MEIS2 in PrCa. However, there remain significant gaps in our understanding of how MEIS proteins suppress tumor progression and the role of HOXB13 in MEIS-mediated tumor suppression.

The function of MEIS proteins is critical but distinct among normal and malignant tissues. Further, the oncogenic vs. tumor-suppressive functions of MEIS proteins depend upon tissue of origin (*Brechka et al., 2017*). MEIS proteins belong to the three amino-acid loop extension (TALE) protein family (*Longobardi et al., 2014*) and are critical for multiple components of normal human development and maintenance, including hematopoiesis (*Argiropoulos et al., 2007*; *Ariki et al., 2014*; *Hisa et al., 2004*), vascular patterning (*Azcoitia et al., 2005*), limb patterning (*Graham, 1994*), and anterior-posterior axis determination in combination with Homeobox (HOX) genes (*Choe et al., 2014*; *Shanmugam et al., 1999*; *Williams et al., 2005*). Increased expression of MEIS proteins is associated with tumorigenesis in certain cancers, including leukemia (*Kumar et al., 2009*), ovarian carcinoma (*Crijns et al., 2007*), and neuroblastoma (*Geerts et al., 2005*). However, in colorectal carcinoma (*Crist et al., 2011*), gastric carcinoma (*Song et al., 2017*), renal cell carcinoma (*Zhu et al., 2017*), and non-small cell lung cancer (*Li et al., 2014*), increased *MEIS* expression is associated with tumor suppression. In some instances, MEIS1 expression results in reduced proliferation by inducing cell cycle arrest at the $G_1/S$ phase transition (*Song et al., 2017*; *Zhu et al., 2017*).

HOX transcription factors play a key role in anterior-posterior axis formation, proliferation, and differentiation (*McGinnis and Krumlauf, 1992*; *Seifert et al., 2015*) but require co-factors to help specify DNA binding (*Mann et al., 2009*), stabilize interactions at the genome level (*Shen et al., 1997a*), and regulate transcription factor activation or repression (*Bürglin, 1998*; *Huang et al., 2005*; *Hyman-Walsh et al., 2010*; *Longobardi et al., 2014*; *Mann et al., 2009*; *Shanmugam et al., 1999*; *Williams et al., 2005*; *Zandvakili and Gebelein, 2016*). Anterior HOX1-8 paralogs prefer to heterotrimerize with MEIS and PBX family proteins (*Ladam and Sagerström, 2014*; *Moens and Selleri, 2006*; *Penkov et al., 2013*; *Slattery et al., 2011*). In the prostate, however, the dominant *HOX* genes expressed are Abd-B-like *HOX* genes and include paralogs 9–13 (*Brechka et al., 2017*; *Huang et al., 2007*). Notably, HOX11–13 paralogs, including HOXB13, prefer to heterodimerize with MEIS1 (*Shen et al., 1997a*) and exclude PBX proteins (*Shen et al., 1997b*). Thus, MEIS/HOX interactions are likely key in prostate development and cancer. Indeed, these combined studies implicate interaction between MEIS1 and the Abd-B-like HOX proteins of the prostate in regulating organ homeostasis. However, the phenotypic impact of MEIS/HOX interactions in PrCa cell gene expression and behavior remains unknown, as do the critical drivers of MEIS/HOX-mediated tumor suppression. Here, we report a phenotypic and mechanistic determination that MEIS proteins promote indolent and non-metastatic prostate cancer via the HOXB13-dependent regulation of extracellular proteoglycans, in particular the multi-RTK inhibitor Decorin. These studies establish critical mechanisms for future utilization of MEIS proteins and predictive biomarkers of indolent prostate cancer and will enable mechanistic studies to define the roles of HOXB13 mutants and oncogenic HOXB13 cofactors in prostate cancer progression.

## Results

### Expression of MEIS1 or MEIS2 in PrCa cells decreases growth in vitro and in vivo

Our previous studies demonstrated that expression of both MEIS1 and MEIS2 is frequently decreased in PrCa patients and that MEIS-positive tumors confer an overall lower risk of biochemical recurrence and metastasis (*Bhanvadia et al., 2018*; *Chen et al., 2012*). Analysis of *MEIS1* and *MEIS2* expression in a panel of PrCa cell lines compared to primary prostate epithelial cell (PrEC) cultures revealed significantly decreased *MEIS1* and *MEIS2* mRNA (p<0.05, *Figure 1A*). Similarly, western blot analysis in all PrCa cell lines documented low protein levels of MEIS1—with the exception of androgen-receptor (AR)-negative lines Du145 and PC3—as well as low levels of MEIS2 (*Figure 1B*). We previously demonstrated that depletion of both MEIS1 and MEIS2 in LAPC4 cells was necessary to promote tumor xenograft growth (*Bhanvadia et al., 2018*). To determine whether increased MEIS expression is sufficient to block PrCa cell growth, we ectopically expressed either MEIS1 or MEIS2 via lentiviral constructs in CWR22Rv1 and LAPC4 PrCa cells (LV-MEIS1 and LV-MEIS2; *Figure 1C*). MEIS2 is known to have several isoforms that differ mainly in the exons used at the C-terminus, as well as one homeodomain-less variant known as MEIS2E (*Figure 1—figure supplement 1A*; *Geerts et al., 2005*). Analyses of mRNA documented that PrECs and prostate cancer cell lines express multiple detectable MEIS2 transcript isoforms, albeit at low levels of expression (*Figure 1*; *Figure 1—figure supplement 1B*). The homeodomain-less and putative dominant-negative MEIS2E isoform was undetectable across multiple lines (*Figure 1—figure supplement 1C*). Further, ectopic expression of either MEIS2A or MEIS2D isoforms are sufficient to inhibit cell growth, while expression of MEIS2E did not impact cell growth (*Figure 1—figure supplement 1D and E*). These data support the necessity of MEIS DNA binding to suppress cell growth. Decreased cell number over time with either MEIS1 or MEIS2A was not associated with increased cell death but was associated with significant accumulation of cells in the $G_1$ phase and fewer cells in $G_2$ (p<0.05, *Figure 1E*; *Figure 1—figure supplement 2*). Subcutaneous xenografting of CWR22Rv1 cells with exogenous MEIS expression into nude mice also resulted in MEIS-mediated tumor suppression in vivo (p<0.05, *Figure 1F&G*). For the ensuing studies we narrowed our analyses to MEIS1 due to the 72.12% sequence similarity shared by MEIS1 and MEIS2, the phenocopied growth suppression in vitro and in vivo (*Figure 1*; *Figure 1—figure supplement 1D and E*), the complexity of experimental design around multiple MEIS2 isoforms, and their similarly reduced expression in prostate tumors and cancer cells (*Bhanvadia et al., 2018*).

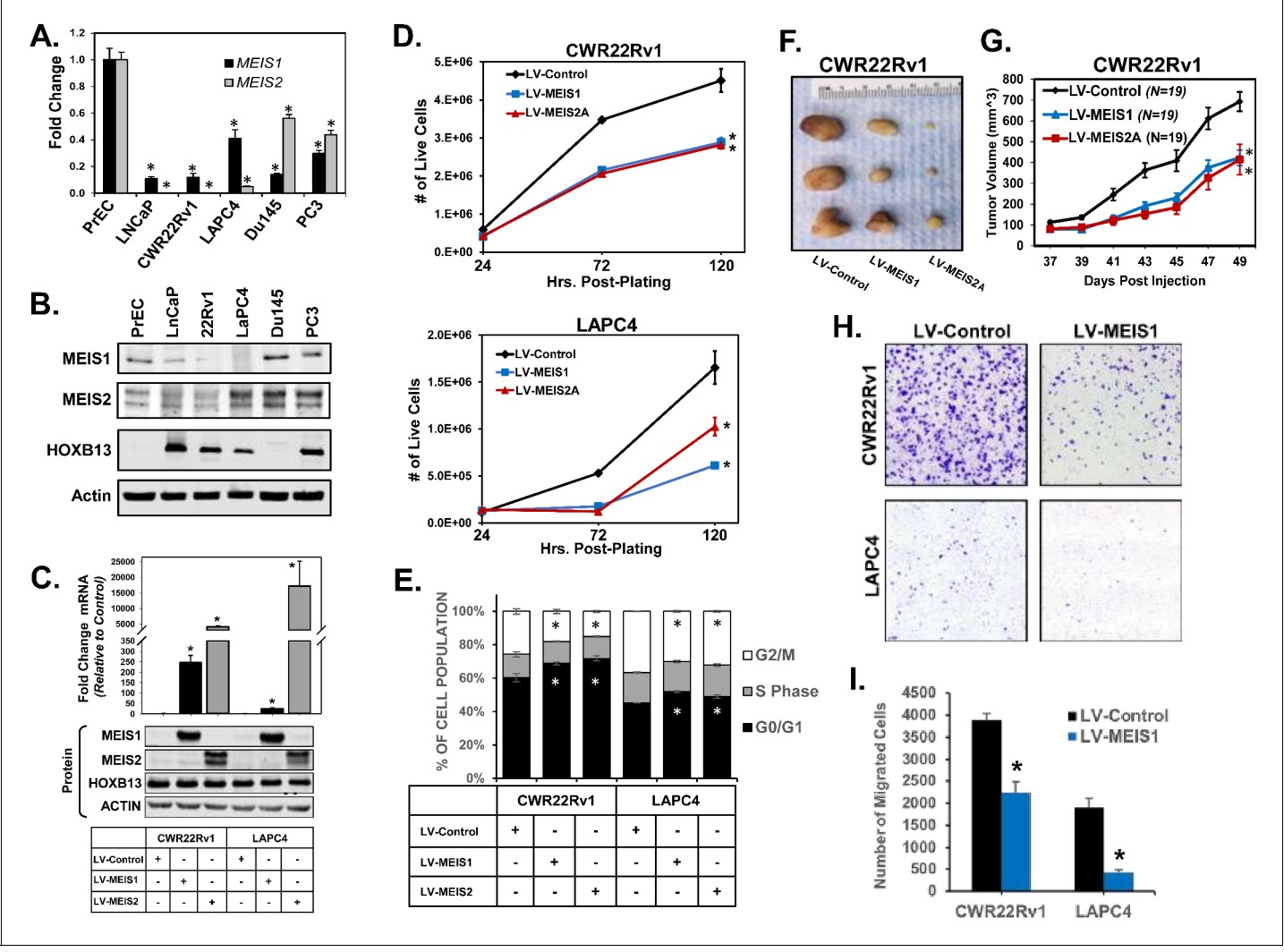

**Figure 1.** Expression of MEIS1 or MEIS2 in PrCa cell lines is sufficient to decrease growth in vitro and in vivo. (**A**) RT-PCR for *MEIS1* (black) or pan-*MEIS2* (gray) in five of the most common prostate cancer cell lines (LNCaP, CWR22Rv1, LAPC4, Du145, and PC3) as compared to non-malignant primary prostate epithelial cells (PrECs). Fold-change from PrEC was calculated using ΔΔCq methodology (technical replicates, n = 3). Error bars represent standard error of the mean (SEM). (**B**) Western blot analysis of endogenous MEIS1, MEIS2, and HOXB13 expression from common prostate cancer cell lines and primary Prostate Epithelial Cell (PrEC) culture. Actin was used as a loading control. (**C**) Western blot confirmation of lentiviral overexpression of MEIS1 or MEIS2 in CWR22Rv1 and LAPC4 cell lines. LV-Control encodes an expression plasmid for constitutive Cas9 expression. Endogenous HOXB13 expression was also assessed in all lines. Actin was used as a loading control. (**D**) Proliferation of CWR22Rv1 (top) and LAPC4 (bottom) with exogenous expression of MEIS1 (blue), MEIS2A (red), or control (black). Cell number over time was assessed by manual counting of live cells on a hemocytometer. Data represent mean count and SEM at each time point (technical replicates, n = 3). Data for LV-Control and LV-MEIS2A is the same as in *Figure 1—figure supplement 2D-E* (**E**) Cell cycle analysis determined by propidium iodide (PI) fluorescence intensity in CWR22Rv1 and LAPC4 cells with exogenous expression of MEIS1 (blue), MEIS2A (red), or control (black). Data represent mean (technical replicates, n = 3) and SEM. (**F**) Representative tumors fixed at time of sacrifice. Tumors are subcutaneous xenografts of CWR22Rv1 cells with exogenous expression of MEIS1, MEIS2A, or control. (**G**) Subcutaneous tumor growth over time for CWR22Rv1 cells with exogenous expression of MEIS1 (blue), MEIS2A (red), or control (black). Data points represent mean (n = 19 for each cell type) and SEM. (**H**) Representative images of transwell migration assays for CWR22Rv1 (top) and LAPC4 (bottom) of cells with exogenous expression of control (left) or MEIS1 (right). (**I**) Quantification of transwell migrations performed in (**H**). Data represent mean (technical replicates, n = 4) and SEM. *Student's t-test p<0.05 for all panels. See also *Figure 1—figure supplements 1* and *2*. The online version of this article includes the following figure supplement(s) for figure 1:

**Figure supplement 1.** MEIS2 Isoforms in Prostate Cancer Cells.
**Figure supplement 2.** Decreased cell number with exogenous MEIS1 or MEIS2 expression is not the result of increased cell death.

Given the correlation between *MEIS* expression and metastasis within annotated tumor specimens, we investigated the migratory capacity of CWR22Rv1 and LAPC4 cells expressing exogenous MEIS1. To minimize the impact of differences in proliferation between LV-MEIS1 and control lines, 3 μM aphidicolin was used to inhibit proliferation in all assays, and is not reported to affect migration (*Müller et al., 2002*). MEIS1-expressing cells showed significantly decreased migration in vitro compared to control cells ($p < 0.05$, *Figure 1H,I*). Thus, both MEIS1 and full-length MEIS2 are sufficient to slow PrCa cell proliferation and tumor growth via reduced $G_1/S$ phase transition and decreased migratory capacity. These data are consistent with findings from other histological tumor types, where MEIS1 functions as a tumor suppressor, slows $G_1/S$ phase transition, and reduces migration and invasion capacity in vitro (*Song et al., 2017*; *Zhu et al., 2017*).

## Exogenous MEIS1 expression rescues the nuclear MEIS1-HOXB13 interaction present in normal PrECs

HOXB13 has critical roles in normal prostate secretory function, differentiation, and response to androgens in rodent prostate models and is implicated in human PrCa (*Chen et al., 2018*; *Economides and Capecchi, 2003*; *Hamid et al., 2014*; *Huang et al., 2007*; *Jung et al., 2004a*; *Jung et al., 2004b*; *Kim et al., 2014a*; *Kim et al., 2010a*; *Kim et al., 2014b*; *Kim et al., 2010b*; *Navarro and Goldstein, 2018*; *Pomerantz et al., 2015*). Comparative analyses of *HOX* gene mRNA expression using publicly-available RNA-Seq datasets of adult human prostate tissues demonstrated that *HOXB13* is the highest-expressed (*Pflueger et al., 2011*; *Robinson et al., 2015*) *HOX* gene across benign epithelium, tumor, and metastatic tissue; *HOXA10* is the next-highest (FPKMs HOXB13 vs. HOXA10: benign, 167.69 vs 38.53; primary tumor, 197.40 vs 35.63; metastasis, 149.44 vs 28.03, *Figure 2A*; *Bhanvadia et al., 2018*). Notably, this result is consistent with observations in both rat and murine prostate, which also have high *HOXA13* and *D13* expression levels though still below the level of *HOXB13* (*Brechka et al., 2017*; *Huang et al., 2007*).

While expression of HOXB13 remains high throughout prostate tumors and metastases, we previously demonstrated a step-wise decrease in *MEIS1* and *MEIS2* expression from benign epithelium to tumor and metastasis and that MEIS-positive prostate tumors confer an overall favorable patient outcome, thus implying that MEIS-HOXB13 interactions are tumor-suppressive (*Bhanvadia et al., 2018*). We thus sought to confirm a nuclear interaction between MEIS1 and HOXB13 within normal PrECs when both proteins are present and in prostate cancer cells when MEIS1 expression is increased. To accomplish this, we used in situ proximity ligation assays (PLA) and co-immunoprecipitation with antibodies specific to MEIS1 and HOXB13 in benign PrECs and in our CWR22Rv1-Control, CWR22Rv1-LV-MEIS1, LAPC4-Control, and LAPC4-LV-MEIS1 cell line models (*Figure 2B*). While PLA does not enable absolute quantitation of interactions within a cell, it does permit quantification of relative differences between samples (*Söderberg et al., 2006*; *Söderberg et al., 2008*; *Weibrecht et al., 2010*). Interactions between MEIS1 and HOXB13 were detectable in normal PrECs; such interactions decreased with lower MEIS expression in prostate cancer cells. Importantly, ectopic MEIS1 expression leading to growth suppression in CWR22Rv1- and LAPC4 PrCa cells was associated with increased MEIS-HOXB13 interactions compared to the respective cell line controls (*Figure 2C*). Furthermore, co-immunoprecipitation of MEIS1 with HOXB13 was observed in PrECs, and increased MEIS1 pulldown was observed in CWR22Rv1 and LAPC4 cells ectopically expressing MEIS1 (*Figure 2D*).

Together, these results document *HOXB13* as the highest-expressed *HOX* gene in the adult human prostate, thus prioritizing the functional importance of HOXB13 in this tissue. These data also suggest that loss of normal MEIS-HOXB13 interactions by decreased MEIS expression could enable non-canonical HOXB13 binding partners to partner with HOXB13 to promote oncogenesis and tumor progression.

## MEIS-mediated suppression of proliferation and migration is dependent upon HOXB13

Having established the tumor-suppressive capability of MEIS1 and its ability to act as a putative HOXB13 cofactor in PrECs, we next sought to test whether MEIS-mediated growth suppression is HOXB13-dependent. To accomplish this, we used CRISPR/Cas9 to knock-out *HOXB13* in CWR22Rv1 and LAPC4 PrCa cells. We then ectopically expressed *MEIS1* in the resulting HOXB13[ko] lines to

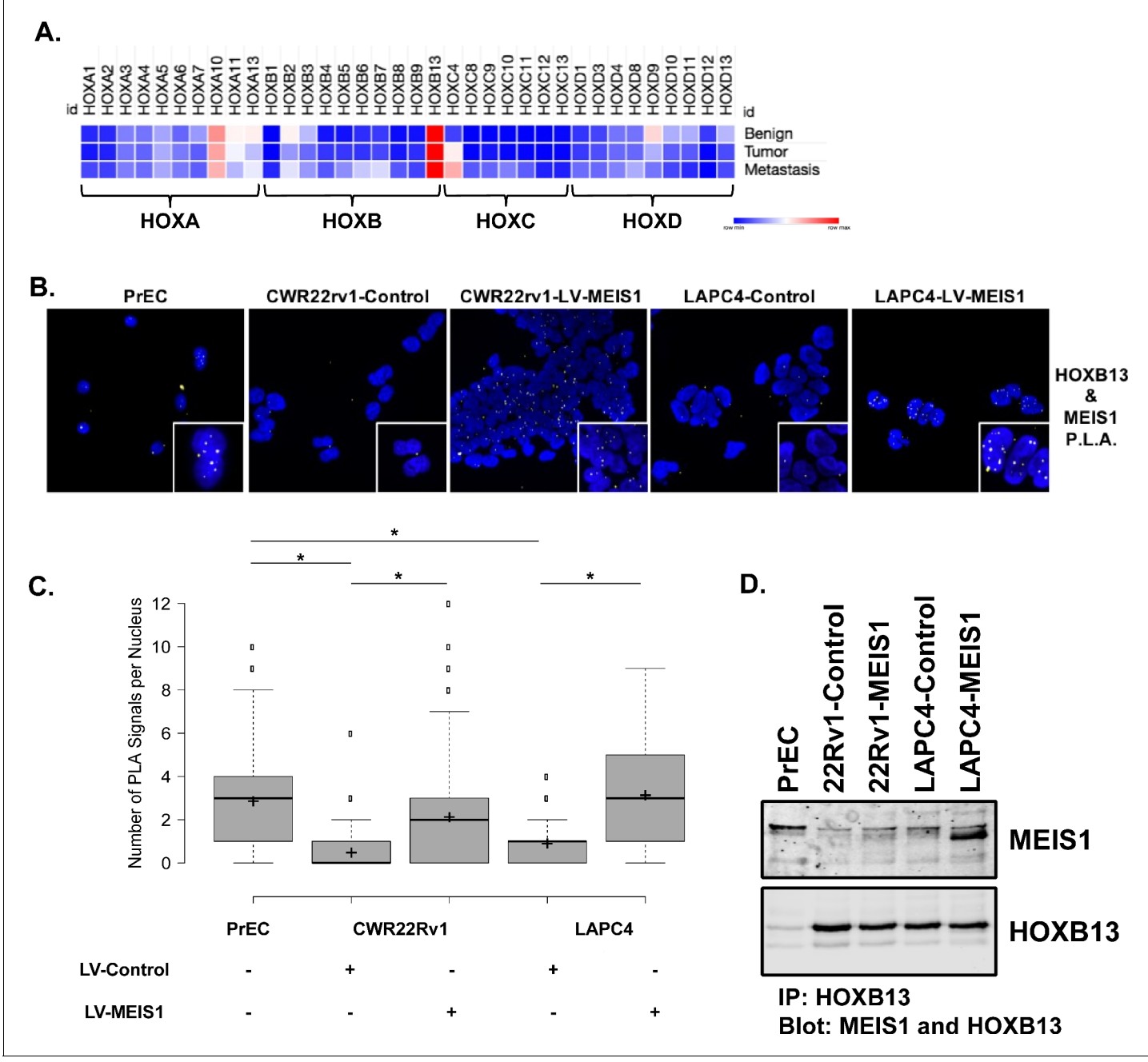

**Figure 2.** Exogenous MEIS1 expression rescues the nuclear MEIS1–HOXB13 interaction present in normal prostate epithelial cells. (**A**) Heatmap for *HOX* expression profile from publicly available, human RNA-seq data of benign prostate, primary prostate tumor, or metastasis from a prostate tumor. Log$_2$(FPKM) values were used to generate heatmap. (**B**) In situ proximity ligation assay (PLA) to identify MEIS1/HOXB13 heterodimers in prostate epithelial cells (PrECs; short-term culture of primary epithelial cells from a de-identified patient, positive for expression of MEIS1 and HOXB13); CWR22rv1-Control (expresses HOXB13 but has low to undetectable expression of MEIS1); CWR22rv1-LV-MEIS1 (expresses HOXB13 and exogenous MEIS1); LAPC4-Control (expresses HOXB13 but has low expression of MEIS1); and LAPC4-LV-MEIS1 (expresses HOXB13 and exogenous MEIS1). Nuclei are stained with DAPI. Yellow puncta are the result of a positive PLA reaction and indicate MEIS1 and HOXB13 are within 40 nm of each other. Puncta were imaged as Texas-red fluorescence and pseudo-colored yellow for better contrast. Cells were imaged on the Keyence BZ-X800 microscope with 60x oil immersion objective. (**C**) Quantification of nuclear PLA signals in (**B**). Center lines show medians; box limits indicate 25$^{th}$ and 75$^{th}$ percentiles as determined by R software; whiskers extend 1.5-times the interquartile range from 25$^{th}$ and 75$^{th}$ percentiles, outliers are represented by dots; crosses represent sample means (n = 141, 422, 726, 107, 114, respectively). Boxplot generated with BoxplotR online tool (*Spitzer et al., 2014*) (*Welch's t-test p<0.05). (**D**) Co-immunoprecipitation of MEIS1 with HOXB13 in PrECs and CWR22Rv1 and LAPC4 Control and LV-MEIS1 expressing cells.

create HOXB13^ko-LV-MEIS1 cells (*Figure 3A*). With HOXB13^ko and HOXB13^ko-LV-MEIS1 lines, we were able to determine the impact of exogenous MEIS1 expression in the absence of HOXB13. Cell proliferation assays demonstrated distinct but agreeable phenotypes between CWR22Rv1 and LAPC4 cell lines. In CWR22Rv1 cells, neither HOXB13^ko alone nor HOXB13^ko-LV-MEIS1 lines significantly differed from control cells, while LV-MEIS1 (with HOXB13 present) remained growth-suppressed (p<0.05, *Figure 3B*). Somewhat surprisingly, in LAPC4 cells, deletion of HOXB13 significantly decreased proliferation to approximately the same rate as the LV-MEIS1 line compared to controls (p<0.05). Further analyses demonstrated that loss of HOXB13 in LAPC4 cells was not associated with increased cell death (*Figure 3—figure supplement 1*). However, in keeping with a requirement for HOXB13 expression with MEIS1 to enact a tumor suppressive effect, ectopic expression of LV-MEIS1 in LAPC4-HOXB13^ko cells significantly *increased* proliferation compared to HOXB13^ko (p<0.05 vs. HOXB13^ko, *Figure 3C*), rather than further decreasing proliferation or

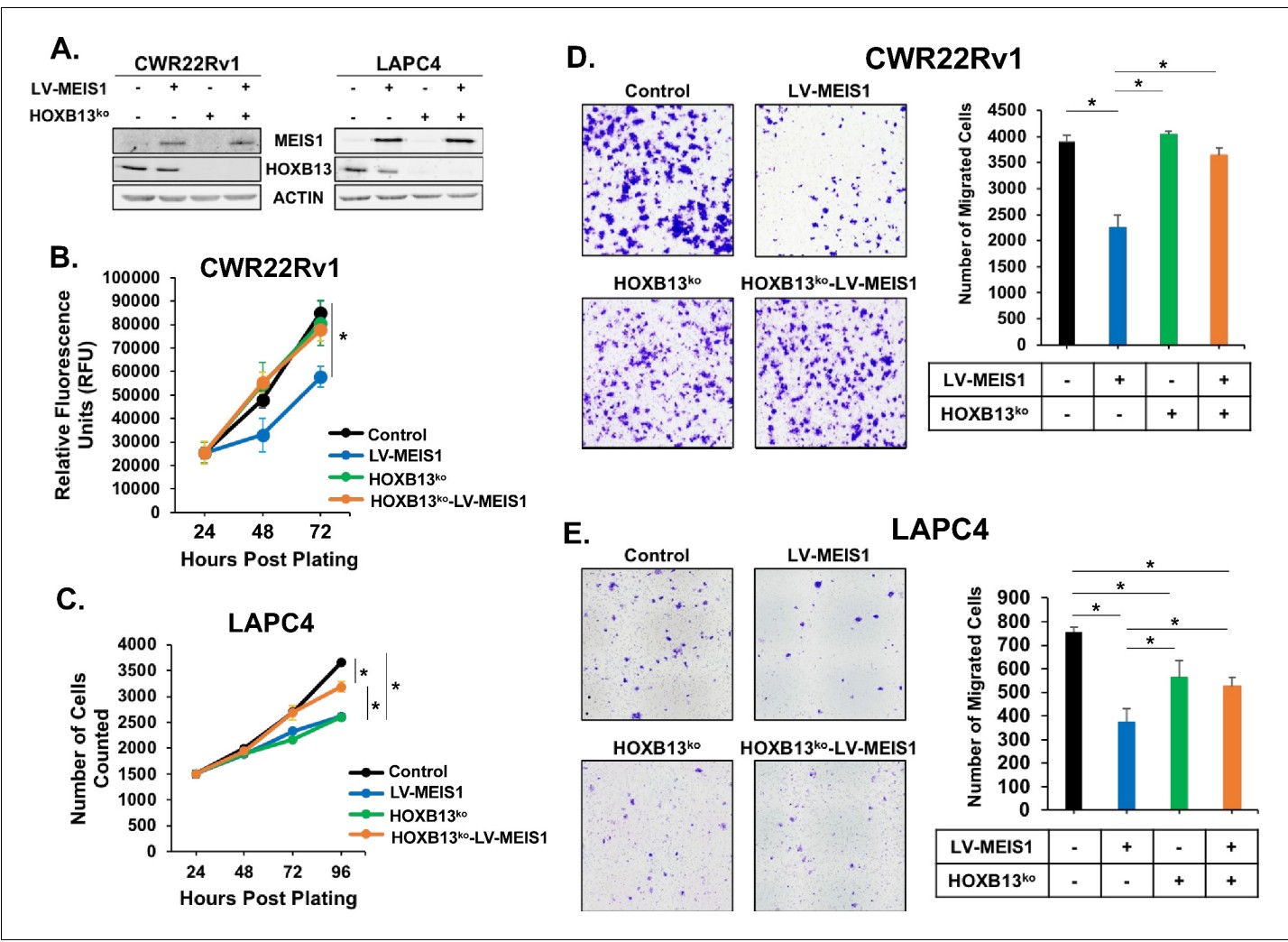

**Figure 3.** MEIS-mediated suppression of proliferation and migration is dependent upon HOXB. (A) Western blot analysis of *HOXB13* knockout using CRISPR and exogenous expression of MEIS1 in the resulting HOXB13^ko cells for CWR22Rv1 and LAPC4. Actin was used as a loading control. (B-C) Proliferation of CWR22Rv1 (B) and LAPC4 (C) cells expressing control (black) or LV-MEIS1 (blue); or with HOXB13^ko (green) or HOXB13^ko and LV-MEIS1 expression (orange). Cell number over time was assessed using CyQuant direct cell proliferation kit. Data represent mean and SEM at each timepoint (technical replicates, n = 3). (D-E) Representative 10x images (left) and quantitation (right) of transwell migration assay for CWR22Rv1 (D) and LAPC4 (E) cell line derivatives. Data represent mean number of cells counted (technical replicates, n = 3) and SEM. *Student's t-test p<0.05 for all panels. See also *Figure 3—figure supplement 1*.

The online version of this article includes the following figure supplement(s) for figure 3:

**Figure supplement 1.** Cell viability is unaffected by HOXB13^ko in LAPC4 cells.

remaining growth suppressed when HOXB13 was present. In this instance, loss of HOXB13 may enable MEIS1 to interact with an alternate HOX protein and differentially regulate cell proliferation. The interaction between MEIS1 and HOXA9, for example, is known to be tumorigenic in leukemia (*Kelly et al., 2011*). However, given the infrequency of HOXB13 loss in prostate tumor specimens, this scenario is not expected to be observed clinically (*Bhanvadia et al., 2018*; *Brechka et al., 2017*).

The HOXB13-dependency of MEIS1 tumor suppressive function in PrCa cells was further demonstrated with the analyses of cell migration phenotypes. In both CWR22Rv1 and LAPC4 cells, HOXB13$^{ko}$ and HOXB13$^{ko}$-LV-MEIS1 lines demonstrated significantly greater migration than LV-MEIS1 cells, while LV-MEIS1 continued to show significantly reduced migration compared to controls (p<0.05, *Figure 3D,E*). Taken together, these results indicate that MEIS-mediated suppression of cell proliferation and migration in PrCa cells requires HOXB13 expression.

## MEIS-mediated metastasis suppression in vivo is HOXB13-dependent

Due to previously published clinical data identifying an increased risk of metastasis with loss of *MEIS1/2* expression (*Bhanvadia et al., 2018*), the results of MEIS-mediated suppression of in vitro migration, and the putative role for HOXB13 in PrCa progression (*Brechka et al., 2017*), we sought to determine the role of MEIS1 expression and/or HOXB13 deletion using a clinically-relevant in vivo metastasis model. We thus performed intracardiac (IC) injection of luciferase-expressing versions of CWR22Rv1-Control, -LV-MEIS1, -HOXB13$^{ko}$, and -HOXB13$^{ko}$-LV-MEIS1 in castrated male athymic nude mice (*Figure 4A*) and monitored metastatic dissemination and growth via in vivo bioluminescent imaging as previously described (*Figure 4B*; *Kregel et al., 2016*). Overall survival post-injection was significantly increased in LV-MEIS1 cells compared to control cells naturally lacking detectable MEIS1 expression (p<0.05). Further, in accordance with our in vitro data, there was no statistically significant difference in overall survival between Control vs. HOXB13$^{ko}$ and HOXB13$^{ko}$-LV-MEIS1 cells (*Figure 4C*). Several clinically relevant organ sites of distant metastases were also observed, including pelvis, vertebrae, skull (maxilla and mandible), kidney, lymph nodes, and lungs (*Figure 4D*). Notably, LV-MEIS1 cells were the only condition in which bone metastases were not observed. These data demonstrate that MEIS1 suppresses prostate tumor growth and metastatic colonization in vivo, and the tumor-suppressive capability of MEIS1 in vivo is dependent upon expression and interaction with HOXB13.

## Integration of ChIP-seq and RNA-seq analyses reveals MEIS1-mediated, HOXB13-dependent, direct regulation of proteoglycans

To identify direct gene targets and pathways regulated by MEIS1 in prostate cells and identify mechanisms of tumor suppression, we performed chromatin immunoprecipitation and sequencing (ChIP-seq) of MEIS1 in the CWR22Rv1-LV-MEIS1 line. Additionally, given the dependence on HOXB13 and to enable determination of HOXB13-dependent vs. HOXB13-independent MEIS1 DNA binding, we performed parallel MEIS1 ChIP-seq in the CWR22Rv1-HOXB13$^{ko}$-LV-MEIS1 line (*Figure 5A*). In the LV-MEIS1 line, where both MEIS1 and HOXB13 are present, we observed 7559 peaks that were annotated to 4161 unique gene targets (*Figure 5A* and *Supplementary file 1*). In the HOXB13$^{ko}$-LV-MEIS1 line, where MEIS1 is present but HOXB13 is absent, we observed only 2048 peaks that were annotated to 1617 unique gene targets (*Figure 5A* and *Supplementary file 2*). The reduction in the number of peaks as well as the shift toward new peak locations of the MEIS1 cistrome in the absence of HOXB13 (*Figure 5A*) are supported by previous literature describing that HOX proteins stabilize MEIS1 on the DNA and that heterodimers of HOX and MEIS proteins develop latent motif specificity that is not demonstrated by either protein individually (*Slattery et al., 2011*).

As further validation of an interaction between MEIS1 and HOXB13, we performed spaced motif (SpaMo) analysis on MEIS1 peaks from both ChIP-seq experiments to identify conserved motifs associating with MEIS1. Unsurprisingly, when both MEIS1 and HOXB13 were present in the LV-MEIS1 line, the HOXB13 motif was significantly conserved in MEIS1 peaks at a distance of 1 bp from the MEIS1 motif itself, strongly supporting a direct interaction between these two proteins (p=2.84×10$^{-12}$, *Figure 5B* top). On the other hand, when HOXB13 was absent, the only HOX motif associated with MEIS1 was HOXA9, which is conserved at a distance of 3 bp from the MEIS1 motif (p=1.14×10$^{-6}$, *Figure 5B* bottom). Notably, due to the highly conserved DNA binding domain of

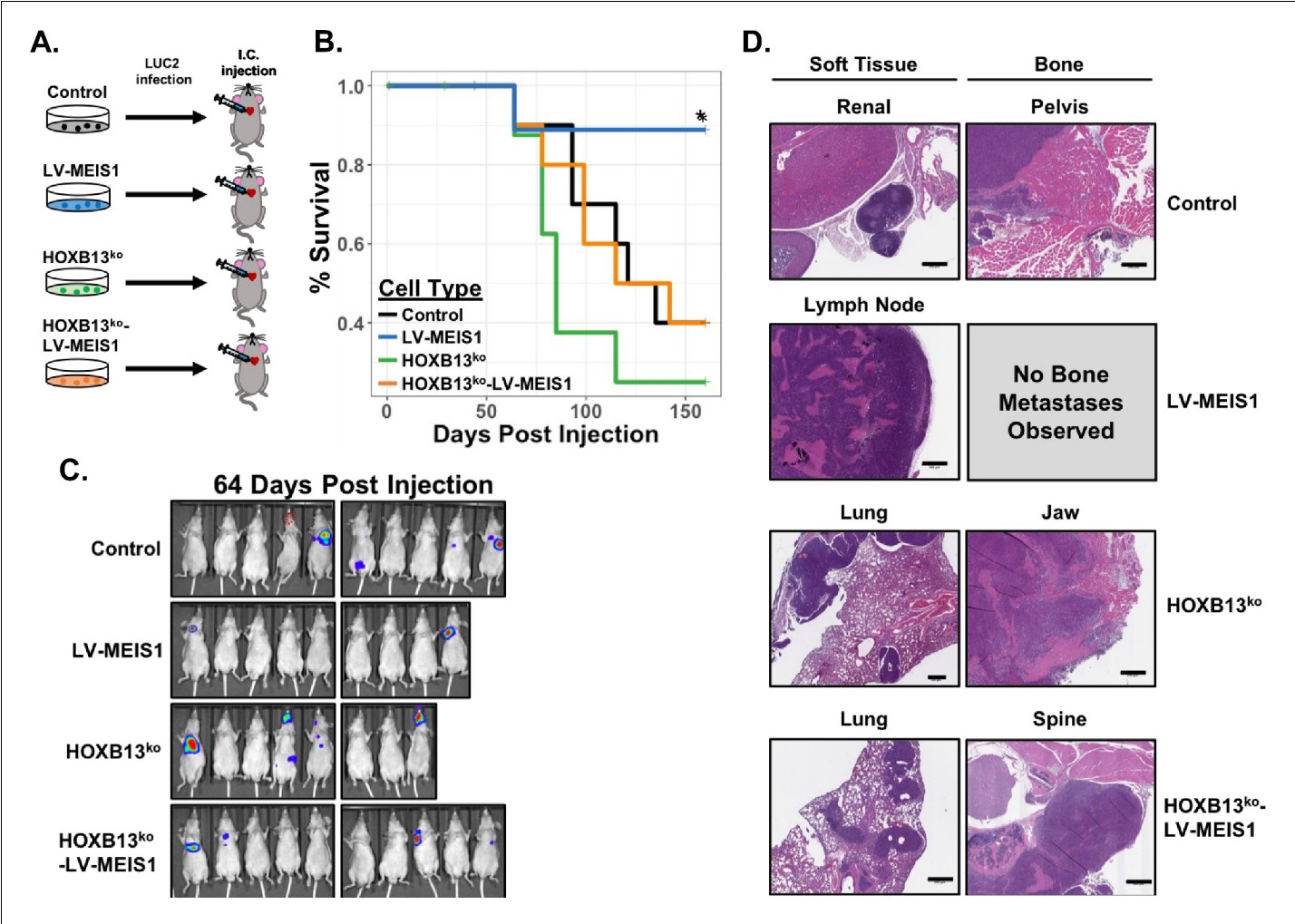

**Figure 4.** MEIS-mediated metastasis suppression in vivo is HOXB13-dependent. (**A**) Schematic of experimental design for intracardiac injection of CWR22Rv1 cell line derivatives into athymic nude mice to model metastasis. Control (black), LV-MEIS1 (blue), HOXB13$^{ko}$ (green), and HOXB13$^{ko}$-LV-MEIS1 (orange) derivatives of CWR22Rv1 were each infected with a lentiviral LUC2 expression vector to enable in vivo bioluminescent monitoring of metastasis formation. LUC2-expressing cells were then injected into the left ventricle of athymic nude mice with 1 cell line per mouse (n = 10 mice per cell line). (**B**) Kaplan-Meier survival curves illustrating overall survival of mice post-intracardiac injection to the veterinarian-approved endpoint. LV-MEIS1 vs. control cells (Chisq = 4.1 on 1 degree of freedom, p=0.04); LV-MEIS1 vs. HOXB13$^{ko}$-LV-MEIS1 cells (Chisq = 4.1 on 1 degree of freedom, p=0.04); LV-MEIS1 vs. HOXB13$^{ko}$ cells (Chisq = 6.1 on 1 degree of freedom, p=0.01); HOXB13$^{ko}$ vs. control cells (Chisq = 1.4 on 1 degree of freedom, p=0.2); HOXB13$^{ko}$ vs. HOXB13$^{ko}$-LV-MEIS1 cells (Chisq = 0.9 on 1 degree of freedom, p=0.3); HOXB13$^{ko}$-LV-MEIS1 vs. control cells (Chisq = 0 on 1 degree of freedom, p=1). (**C**) Representative images of in vivo bioluminescent imaging of the metastatic colonization of CWR22Rv1 intracardiac-injected mice. Red dashed circle in the control mouse image indicates location of a visually evident and palpable metastasis that did not produce bioluminescent signal. (**D**) Representative H and E images (20 × magnification) of histological sections of metastases from injected mice. Scale bars indicate 500 μm.

many HOX genes, the HOXA9 motif potentially associated with MEIS1 in the absence of HOXB13 also showed significant similarity to the Abd-B-like HOX general core motif (E-value = $1.3373 \times 10^{-6}$) and PBX1 motif (E-value = $1.4640 \times 10^{-6}$) (*Figure 5—figure supplement 1A*). Thus, it is feasible that MEIS1 associates with HOXA10 in the absence of HOXB13, since HOXA10 is the next-highest-expressed HOX in the prostate (*Figure 2A*).

We next conducted RNA-seq to identify MEIS1-mediated gene regulation. These analyses compared global gene expression of CWR22Rv1-Control and CWR22Rv1-LV-MEIS1 as well as HOXB13$^{ko}$ and HOXB13$^{ko}$-LV-MEIS1 cells to precisely delineate MEIS1- and HOXB13-regulated genes (*Figure 5—figure supplement 1B*). Importantly, inclusion of HOXB13$^{ko}$ lines enabled determination of HOXB13-associated gene regulation as well as identification of significant changes between LV-MEIS1 and control cells that were HOXB13-independent and thus unrelated to tumor suppression

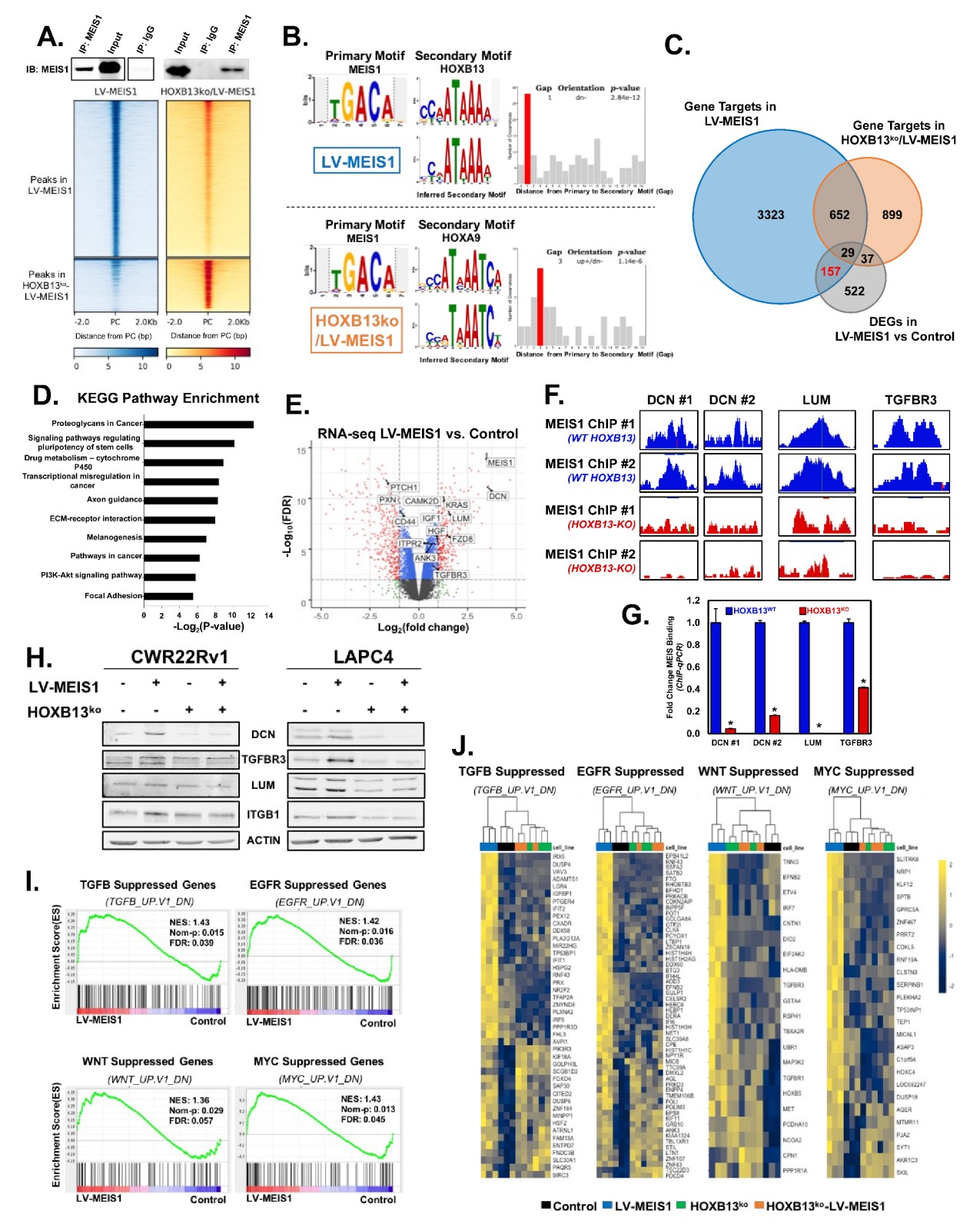

**Figure 5.** Integration of ChIP-seq and RNA-seq analyses reveals MEIS1-mediated, HOXB13-dependent, direct regulation of proteoglycans including Decorin (DCN). (A) Western blot confirmation of pull-down of MEIS1 from chromatin in CWR22Rv1 LV-MEIS1 and HOXB13ko-LV-MEIS1 cell lines (Top) and heatmap of read density profiles from MEIS1 ChIP-seq for ±2 kb of the peak center (PC) called by MACS2 (Bottom). Graphs are separated vertically into peaks called from LV-MEIS1 samples vs. input DNA (LV-MEIS1 peaks), where both MEIS1 and HOXB13 are expressed, and peaks called from

*Figure 5 continued on next page*

Figure 5 continued

HOXB13[ko]-LV-MEIS1 samples vs. input DNA (HOXB13[ko]-LV-MEIS1 peaks), where only MEIS1 is present and HOXB13 has been knocked-out. (B) Spaced Motif (SpaMo) analysis querying MEIS1 as the primary motif demonstrates an inferred secondary motif matching HOXB13 has a conserved spacing from the MEIS1 motif that is most significant at 1 bp away, downstream and on the opposite DNA strand in MEIS1 peaks from LV-MEIS1 ChIP-seq (top). In MEIS1 peaks from HOXB13[ko]-LV-MEIS1 ChIP-seq, SpaMo analysis identifies an inferred motif matching HOXA9 as the only HOX motif with a significant conservation of spacing with the MEIS1 motif and is most significant at 3 bp away on either DNA strand regardless of up or downstream (bottom). (C) Venn diagram demonstrating the number of genes with differential expression (fold-change >±1.5, FDR < 0.05) in CWR22Rv1-LV-MEIS1 vs. CWR22Rv1-Control cells from RNA-seq of these lines (gray), overlapped with the number of genes that MEIS1 peaks were annotated to using HOMER in either LV-MEIS1 (blue) or HOXB13[ko]-LV-MEIS1 (orange) ChIP-seq. The 157 genes (red) represent prioritized MEIS effector genes, since they are bound and differentially expressed only when HOXB13 is present. (D) Top 10 significantly enriched pathways from KEGG pathway enrichment analysis using the list of 157 genes identified in (C). (E) Volcano plot of gene expression $Log_2$(fold-change) vs. significance (FDR) in CWR22Rv1 LV-MEIS1 vs. control cells. Dashed lines indicate thresholds for fold-change >2 or<2 and for FDR value <0.01. Highlighted gene symbols indicate genes that: 1) are within the 'proteoglycans in cancer' pathway curated by KEGG, 2) are annotated as targets in the MEIS1 ChIP-seq from LV-MEIS1 cells, and 3) have significant differential expression in LV-MEIS1 vs. control RNA-seq (fold-change >±1.5, FDR < 0.05). Gray dots indicate genes with no significant differential expression; green dots indicate genes with fold-change >±2 but FDR > 0.01; blue dots indicate genes with FDR < 0.01 but fold-change <±2; and red dots indicate genes with both FDR < 0.01 and fold-change >±2. (F) Integrated Genome Browser tracks of MEIS1 ChIP in the presence (WT-HOXB13, blue) and absence of HOXB13 (HOXB13-KO, red) at the DCN (2 regions, DCN #1 and DCN #2), LUM, and TFGBR3 loci. (G) ChIP-qPCR of MEIS1 binding using site-specific genomic primers against DCN, LUM, and TGFBR3 loci. MEIS1 binding is significantly diminished when HOXB13 is deleted (HOXB13KO; * indicates p<0.05). (H) Western blot analysis of key genes with significant differential expression in RNA-seq between CWR22Rv1-LV-MEIS1 and control. DCN, TGFBR3, and LUM are direct targets of MEIS1 from ChIP-seq data and ITGB1 is a downstream target of LUM. (I) Gene set enrichment analysis (GSEA) from RNA-seq between CWR22Rv1-LV-MEIS1 and control cells on the oncogenic signatures collection from MSigDB. Enrichment is observed in CWR22Rv-LV-MEIS1 cells for genes known to be suppressed by: active TGFβ signaling (TGFB_UP.V1_DN, NES: 1.43, FDR: 0.039), active EGFR signaling (EGFR_UP.V1_DN, NES: 1.42, FDR: 0.036), active WNT signaling (WNT_UP.V1_DN, NES:1.36, FDR 0.057), and active c-MYC signaling (MYC_UP.V1_DN, NES: 1.43, FDR: 0.045). (J) Heatmaps of expression and unsupervised clustering of RNA-seq from CWR22Rv1 control (purple), LV-MEIS1 (blue), HOXB13[ko] (green), and HOXB13[ko]-LV-MEIS1 (orange) cells for all genes in the leading edge of enrichment for each gene set in (G). See also *Figure 5—figure supplement 1* and *Supplementary files 1–6*.

The online version of this article includes the following figure supplement(s) for figure 5:

**Figure supplement 1.** HOXA9 motif shares similarity, RNA-seq MDS plot, and Validation of DCN expression in VCaP.
**Figure supplement 2.** Functional impact of MEIS1 and DCN knockdown in LAPC4.

(*Supplementary file 3*). Gene regulation was defined as direct MEIS1 binding via ChIP, increased mRNA expression when MEIS1 was ectopically expressed, and loss of MEIS1 genome binding and mRNA expression when MEIS1 was expressed but HOXB13 was deleted. Integration of RNA-seq and ChIP-seq data revealed 745 differentially expressed genes (DEGs) (edgeR, fold-change >1.5 and FDR < 0.05) in CWR22Rv1-LV-MEIS1 compared to controls (*Supplementary file 4*). Of those 745 DEGs, 186 were directly targeted by MEIS1; of these targets, 29 genes were also bound by MEIS1 in the HOXB13[ko] condition and were thus removed since they would not be expected to be critical mediators of MEIS1–HOXB13-mediated tumor suppression. The resulting 157 DEGs (*Supplementary file 5*) represent genes that are direct targets of MEIS1 only when HOXB13 is present and therefore represent prioritized candidates to elucidate the mechanism of MEIS1-dependent tumor suppression (*Figure 5C*). Pathway analyses of these 157 genes prioritized multiple putative pathways, of which 'proteoglycans in cancer' was the most enriched pathway associated with MEIS1 and HOXB13 expression (*Figure 5D*). Further analyses documented that the majority of proteoglycans targeted by MEIS1 were upregulated by MEIS1 expression (*Figure 5E*).

Of particular interest was elevated expression of DCN, which was one of the most increased of the significant DEGs in the dataset (fold-change = 11.38, FDR = $7.37 \times 10^{-12}$). DCN belongs to the small-leucine-rich-proteoglycan (SLRP) family of proteins that has been well-documented to decrease tumor growth and progression (*Bi and Yang, 2013*; *Csordás et al., 2000*; *Edwards, 2012*; *Goldoni et al., 2009*; *Hildebrand et al., 1994*; *Iozzo et al., 1999*; *Järvinen and Prince, 2015*; *Khan et al., 2011*; *Santra et al., 2002*; *Schönherr et al., 1998*; *Schönherr et al., 2005*; *Zhang et al., 2018*; *Zhu et al., 2005*). Lumican (LUM), which also increased with LV-MEIS1 expression (fold-change = 2.88, FDR = $1.79 \times 10^{-9}$), is another member of the tumor-suppressive SLRP protein family and has been shown to increase integrin B1 (ITGB1)-mediated adhesion as well as regulate expression of ITGB1 (*D'Onofrio et al., 2008*; *Jeanne et al., 2017*; *Zeltz et al., 2010*). In parallel, TGFBR3 (also known as betaglycan) significantly increased with MEIS1 expression (fold-change = 1.67, FDR = $4.13 \times 10^{-4}$) and has been shown to inhibit TGFβ signaling and decrease

prostate tumor growth and progression in a manner similar to DCN (*Ajiboye et al., 2010*; *Eickelberg et al., 2002*; *Sharifi et al., 2007*; *Turley et al., 2007*). Analyses of MEIS1 ChIP-Seq demonstrated binding in the *DCN*, *LUM*, and *TGFBR1* genomic region (*Figure 5F*), and independent ChIP-qPCR validated MEIS1 binding (*Figure 5G*). Importantly, MEIS1 binding was significantly diminished when HOXB13 was deleted (*Figure 5F and G*). Increased mRNA expression of *DCN, TGFBR3, LUM,* and *ITGB1* were validated at the protein level in both CWR22Rv1 and LAPC4 cell lines (*Figure 5H*). Additionally, increased protein expression was not observed when HOXB13 was absent, thus verifying the dependency of HOXB13 interaction to regulate expression of these targets. The observed MEIS-mediated increase in DCN mRNA and protein expression was also observed in a third prostate cancer cell line, VCAP (*Figure 5—figure supplement 1C*). Moreover, we previously demonstrated that dual MEIS1/MEIS2 knockdown in LAPC4 cells increased tumor xenograft growth *Bhanvadia et al., 2018*; analysis of DCN protein in these cells showed decreased DCN expression when MEIS1, MEIS2, and both MEIS1 and MEIS2 were depleted using shRNAs (*Figure 5—figure supplement 2A*).

DCN is a multi-RTK inhibitor and likely has the broadest functional impact of these proteoglycans on cancer-associated pathways and response to growth factors. The most well-established role for DCN is as an inhibitor of TGFβ signaling (*Baghy et al., 2012*; *Harper et al., 1994*; *Yamaguchi et al., 1990*; *Zhu et al., 2007*). DCN can also exert tumor-suppressive functions via affecting multiple other signaling pathways, including EGFR, IGFR1, AKT, and cMYC. DCN inhibits EGFR signaling after transient activation, leading to increased p21 expression (*Csordás et al., 2000*; *Hu et al., 2009*; *Moscatello et al., 1998*; *Santra et al., 1997*; *Seidler et al., 2006*), and DCN binds and inhibits IGF1R and downstream AKT signaling in cancer cells (*Iozzo et al., 2011*; *Morrione et al., 2013*; *Schönherr et al., 2005*). DCN also antagonizes the c-MET receptor, which can lead to decreased non-canonical β-catenin and decreased cMYC (by way of increased phospho-T58, which destabilizes cMYC and leads to degradation) (*Goldoni et al., 2009*). To test activity of these various pathways, we used MSigDb curated gene sets for oncogenic signatures to perform gene set enrichment analysis (GSEA). The results in *Figure 5I* indicate that the LV-MEIS1 condition, where DCN expression is high, has significantly decreased pathway activation for *TGFβ*, (FDR: 0.039), *EGFR* (FDR: 0.036), *WNT* (FDR: 0.057), and *MYC* (FDR: 0.045) gene sets. These four specific gene sets represent genes normally suppressed by oncogene activation, and their enrichment in the LV-MEIS1 condition suggests decreased activity of the specified oncogenic pathway. This also correlates with increased DCN expression as an established inhibitor of these oncogenic pathways.

We then used RNA-seq data from the four CWR22Rv1 cell line variants (control, LV-MEIS1, HOXB13$^{ko}$, and HOXB13$^{ko}$-LV-MEIS1) to conduct a leading edge of enrichment analysis to further solidify that decreased activity of these oncogenic pathways is a direct result of regulation by MEIS1 and HOXB13 (*Figure 5J*). GSEA for LV-MEIS1 vs. control was also performed on the MSigDb curated gene sets for 'gene ontology: biological processes', which further supported and expanded our findings with enrichment in pathways including: regulation of epithelial to mesenchymal transition, growth factor binding, integrin mediated signaling, and regulation of response to transforming growth factor beta stimulus (*Supplementary file 6*). These data prioritize proteoglycan-mediated tumor suppression, particularly DCN expression, as key mediators of MEIS1-HOXB13-induced tumor suppression in PrCa cells.

## Knockdown of DCN partially reverses MEIS-mediated tumor suppression

Given the potential role of *DCN* as a critical gene target of MEIS1–HOXB13-mediated tumor suppression, we sought to functionally validate the ability of DCN to regulate tumor suppression in MEIS1-expressing CWR22Rv1 and LAPC4 PrCa cells. We thus depleted DCN expression using an siRNA pool to knock-down DCN in LV-MEIS1 lines (*Figure 6A*). In comparison with the decreased proliferation observed with LV-MEIS1 expression, knockdown of DCN partially abrogated the growth suppression by LV-MEIS1 in CWR22v1 cells (p<0.05) but did not have a significant effect in LAPC4 cells (*Figure 6B,C*). However, siRNA knockdown of DCN in both LV-MEIS1 lines was sufficient to partially restore migratory capacity (p<0.05, *Figure 6D,E*).

We also investigated the effect of DCN knockdown on some of the previously identified pathways under DCN control (*Figure 5G*). Western blot analyses revealed increased DCN in the presence of MEIS1, with concomitant decreases in EGFR signaling (decreased phospho-EGFR and increased total

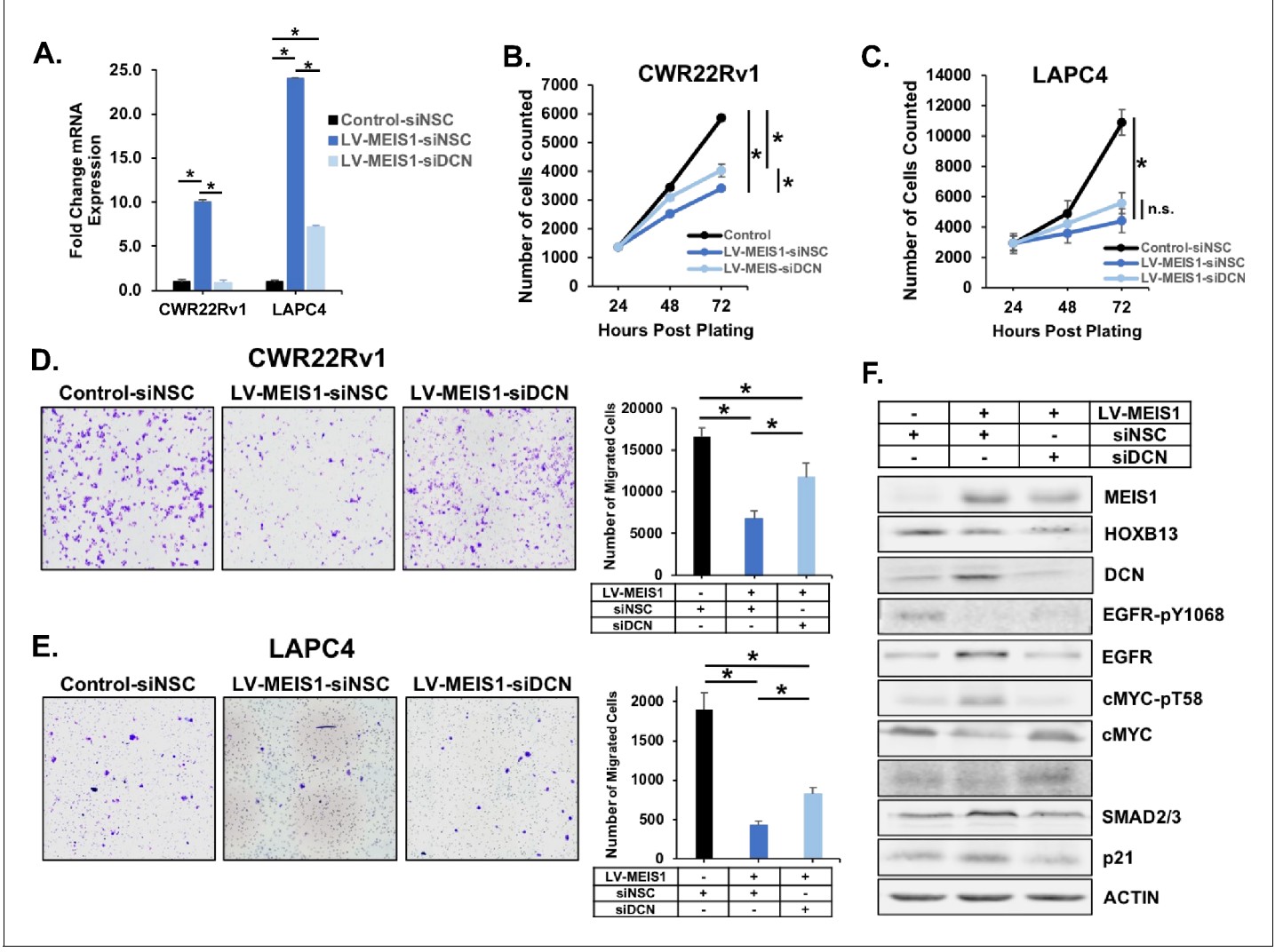

**Figure 6.** Knockdown of *DCN* partially reverses MEIS-mediated tumor suppression. (**A**) RT-PCR in both CWR22Rv1 and LAPC4 for *DCN* expression in control (black) and LV-MEIS1 (dark blue) cells treated with a non-silencing control siRNA pool (siNSC, 4 siRNAs in pool) or LV-MEIS1 cells treated with an siRNA pool targeting *DCN* (light blue) (siDCN, 4 siRNAs in pool). Fold-change from control-siNSC was calculated using ΔΔCq methodology (technical replicates, n = 3). *Student's t-test p<0.05. (**B-C**) Proliferation of CWR22Rv1 (**B**) and LAPC4 (**C**) cell lines from (**A**) was assessed using CyQuant direct cell proliferation assay. Data represent mean and SEM at each timepoint (technical replicates, n = 3). (**D-E**) Representative 10X images (left) and quantitation (right) of transwell migration assay for CWR22rv1 (**D**) and LAPC4 (**E**) cell line derivatives (control-siNSC, LV-MEIS1-siNSC, LV-MEIS1-siDCN). Data represent mean and SEM (technical replicates, n = 3). (**F**) Western blot analysis of MEIS1, HOXB13, DCN, and known downstream targets of DCN in CWR22Rv1 control-siNSC, LV-MEIS1-siNSC, and LV-MEIS1-siDCN. *Student's t-test p<0.05 for all panels. See also *Figure 5—figure supplement 2*. The online version of this article includes the following figure supplement(s) for figure 6:

**Figure supplement 1.** Western blot analysis of MEIS1, HOXB13, DCN, and known downstream targets of DCN in LAPC4 control-siNSC, LV-MEIS1-siNSC, and LV-MEIS1-siDCN cells.

EGFR), increased p21, decreased cMYC signaling (decreased total cMYC and increased degradation signal at phospho-T58), and decreased TGFβ signaling (decreased phospho-SMAD2 and increased total SMAD2/3) (*Figure 6F*). Knockdown of DCN in MEIS-expressing cells resulted in partial or complete restoration of EGFR activation, cMYC expression, SMAD2 activation, and decreased p21 expression. Concordant results were also observed in LAPC4 cells with DCN knockdown (*Figure 6— figure supplement 1*). The ability of DCN knockdown to restore the activity of EGFR, MYC, and TGFβ pathways, which decreased as a result of LV-MEIS1 expression, along with the partial restoration of cellular migration and proliferation in CWR22Rv1 cells with DCN knockdown, strongly

supports a critical role for DCN as a key mediator of MEIS1/HOXB13-dependent metastasis suppression.

### Increased proteoglycan expression within human prostate tumors that retain MEIS1/2 expression

Previous analysis of DCN in prostate tissues indicated high stromal expression compared to epithelial staining (*Henry et al., 2018*). Protein analyses of DCN, MEIS1, and HOXB13 in primary PrECs indicated expression of both MEIS1 and HOXB13 and detectable DCN expression (*Figure 7A*). We subsequently analyzed publicly available RNA-seq datasets from human prostate tumors for associations between MEIS- and DCN-regulated pathways (*Abeshouse et al., 2015*). These analyses of patient-derived datasets agree with our cell line data, whereby *MEIS* expression demonstrated a positive correlation (p<0.05) with *DCN, TGFBR3, LUM*, and *ITGB1* (*Figure 7B*).

## Discussion

PrCa remains the second leading cause of cancer-related death among men in the United States. As such, the continued effort to identify novel gene networks that regulate aggressiveness of PrCa is

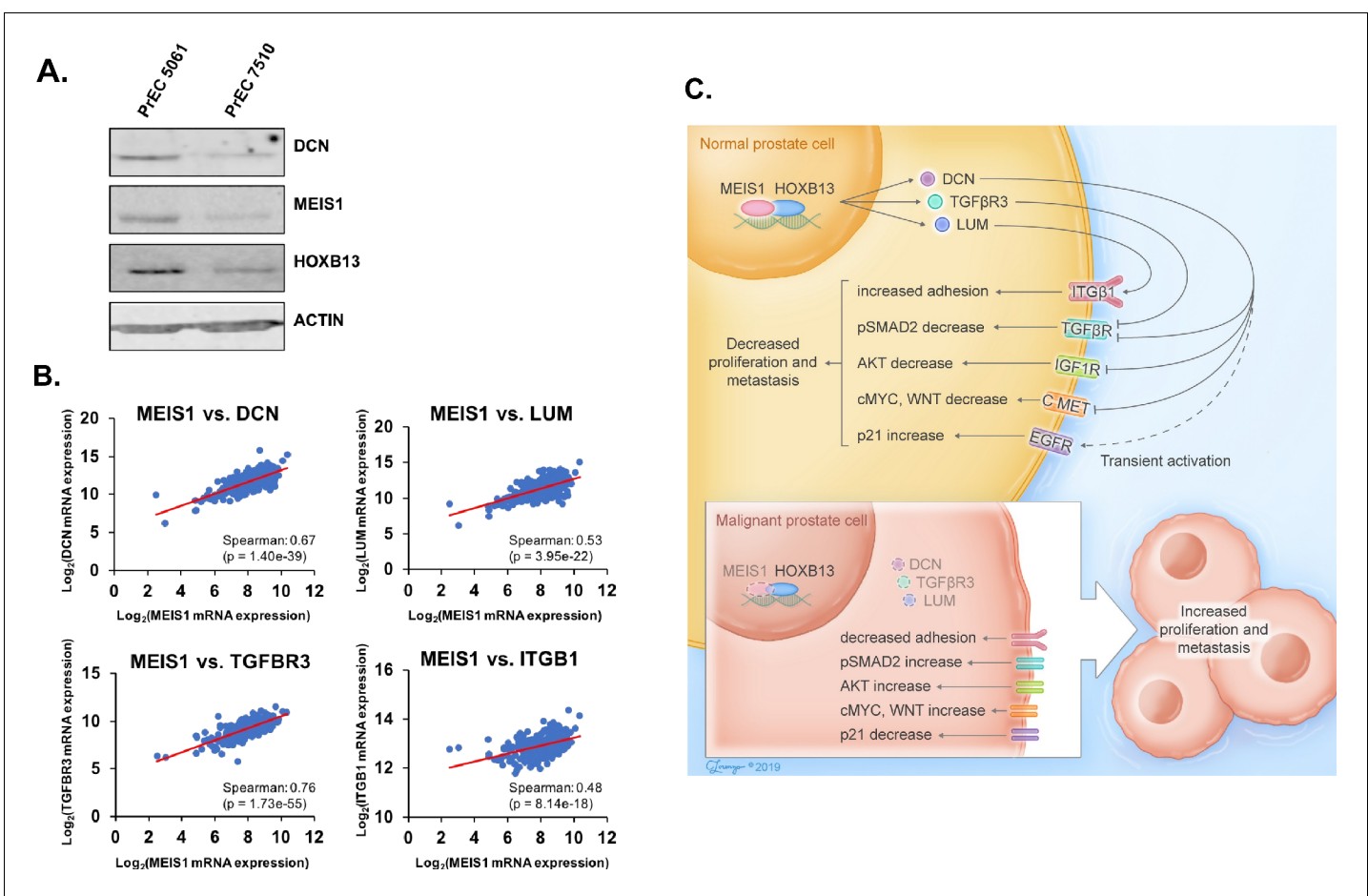

**Figure 7.** Increased proteoglycan expression within human prostate tumors that retain MEIS1/2 expression. (**A**) Western blot analysis of MEIS1 and DCN expression in protein lysates from primary prostate epithelial cells (PrECs). Actin was used as a loading control. (**B**) Correlation of mRNA expression level of *MEIS1* with *DCN, TGFBR3, LUM*, or *ITGB1* in RNA-seq data of prostate tumors from The Cancer Genome Atlas (TCGA; RNA-seq V2 RSEM) (*Abeshouse et al., 2015*). Analysis was performed using the online tool cBioPortal (*Cerami et al., 2012*). (**C**) Schematic of Results. Within normal prostate epithelial cells, MEIS1 complexes with HOXB13 to maintain expression of proteoglycans such as DCN, LUM, and TGFBR3 and repress growth factor and migration/invasion signaling through RTKs. As a cell transforms to a malignant state, *MEIS1/2* are epigenetically silenced in more aggressive prostate tumors and expression of tumor-suppressive proteoglycans is suppressed, leading to decreased regulation of oncogenic signaling through pathways such as TGFβ, EGFR, cMYC, WNT, and IGF1R.

crucial to develop predictive biomarkers that prioritize patients for more aggressive therapies as well as identify novel pharmacological targets. Prior studies implicated MEIS1 and MEIS2 as putative tumor suppressors in PrCa based upon clinical associations between retention of tumor expression and overall survival or risk of metastasis (*Bhanvadia et al., 2018*; *Chen et al., 2012*). However, these data have remained largely unexploited due to a lack of mechanistic understanding of pathways dysregulated by MEIS1/2. In this study, we describe that MEIS1—cooperatively with HOXB13—is responsible for maintaining expression of secreted, tumor-suppressive proteoglycans such as DCN. Without expression of these proteoglycans, there is decreased regulation of several established and potent oncogenic pathways such as EGFR, TGFβ, MYC, and c-MET that can lead to cancer growth and metastasis (*Figure 7C*). Therefore, in addition to the prognostic utility of MEIS1/2 expression in PrCa, we report a mechanism of MEIS-driven tumor suppression that has potential to be exploited clinically.

Decreased expression of secreted proteoglycans in MEIS-low prostate tumors implies an interesting opportunity for treatment via re-administration of these proteins to the tumor microenvironment. Importantly, DCN is viewed as a nontoxic, natural biological product and thus less likely to be immunogenic by itself when administered to patients (*Pucci-Minafra et al., 2008*; *Sofeu Feugaing et al., 2013*). Numerous reports underscore the potential effectiveness of DCN as a therapeutic agent across multiple cancer types. Notably, systemic delivery of recombinant DCN core protein in a prostate-specific $Pten^{-/-}$ mouse cancer model can slow tumor growth and progression compared to a saline control (*Hu et al., 2009*). In breast cancer, both systemic delivery of DCN core protein and intratumoral injection of adenovirus encoding for DCN can decrease growth and metastasis in xenograft mouse models (*Goldoni et al., 2008*; *Tralhão et al., 2003*). Similarly, osteosarcoma cells stably expressing DCN and injected subcutaneously to the backs of mice show significantly fewer pulmonary metastases compared to controls (*Shintani et al., 2008*). These previous studies, while all performed in pre-clinical models, combined with our findings presented here, emphasize the potential utility of DCN and other SLRPs as potent therapies to block cancer metastasis. Thus, further investigation into the optimal production and delivery methods for use in humans are warranted.

Ewing et al. reported the presence of germline $HOXB13^{G84E}$ mutations in a cohort of men with strong family histories of PrCa (*Ewing et al., 2012*). This particular mutation significantly increases a male carrier's risk of being diagnosed with PrCa and also having early-onset disease, higher PSA, and higher Gleason grade at time of diagnosis (*Brechka et al., 2017*; *Zhang et al., 2016*). The mechanism(s) surrounding such increased predisposition to PrCa remains unclear, but this mutation and the $HOXB13^{G135E}$ mutation discovered in Chinese men occur within the MEIS-interacting domains of HOXB13 (*Ewing et al., 2012*; *Lin et al., 2013*). Our findings that the MEIS1–HOXB13 complex is crucial to maintain the tumor-suppressing function of MEIS1 portray these mutations in a new light. Further, the $HOXB13^{G84E}$ mutation is associated with an increased odds ratio (OR) for colorectal cancer (OR: 2.8, p=0.02), bladder cancer (OR: 1.99, p=0.06), and leukemia (OR: 3.17, p=0.01), further demonstrating the importance of the need to understand how $HOXB13$ mutations promote cancer (*Akbari et al., 2013*; *Beebe-Dimmer et al., 2015*). A recent study did not detect an effect of the G84E mutation on interaction with MEIS1 (*Johng et al., 2019*), suggesting alternative mechanisms for HOXB13 mutations in driving prostate carcinogenesis beyond disruption of MEIS interaction. Thus, further research is needed to define the transcriptional impact of $HOXB13$ mutations on MEIS interaction and the regulation of MEIS–HOXB13-associated gene targets.

The interdependence of MEIS1 and HOXB13 to promote tumor suppression implies that changes to either factor enable the other factor to pair with an oncogenic driver. Recent reports demonstrate an affinity between HOXB13 and AR, specifically within malignant cells, and such an interaction potentially contributes to castration resistance (*Chen et al., 2018*; *Navarro and Goldstein, 2018*; *Pomerantz et al., 2015*). Moreover, changes to MEIS/HOXB13 transcriptional regulation or interaction by $HOXB13$ mutations could enable MEIS proteins to pair with HOXA9 or HOXA10 to drive oncogenesis and progression, such as in leukemia or ovarian cancer (*Kelly et al., 2011*; *Kroon et al., 1998*). This model is supported by our data in which HOXB13^ko led to increased proliferation of LAPC4 cells expressing MEIS1 when HOXB13 was deleted, and MEIS1 paired with a HOXA9 motif when HOXB13 was absent. Collectively, these data purport a model whereby epigenetic silencing of MEIS1/2 in prostate tumors allows oncogenic AR/HOXB13 interactions; HOXB13 mutations could modify the affinity of HOXB13 for AR vs. MEIS, diminish MEIS/HOXB13

transcriptional regulation of tumor suppressive genes such as DCN, or enable increased oncogenic MEIS-HOXA9/10 interactions.

Our study provides a compelling argument for regulation of proteoglycans by MEIS1 and HOXB13 as a mechanism for MEIS-driven tumor suppression in PrCa. Further, our findings are supported by previous reports of MEIS1 slowing growth of cancer cells through a $G_1$ cell cycle blockade and reduced migration. However, this study is not without limitations. First, we cannot exclude the contribution of non-proteoglycan pathways to MEIS-mediated tumor suppression. This is exemplified by restoration of cell growth and migration only partially with DCN knockdown in combination with exogenous MEIS1 expression. Second, there may be MEIS2-specific functions that contribute to tumor suppression. This is unlikely, however, given the phenocopied tumor suppression phenotype, high degree of homology between MEIS1 and MEIS2 (72.12% similarity), and previously published data demonstrating that knockdown of both MEIS1 and MEIS2 is required for increased PrCa cell line aggression in vivo (*Bhanvadia et al., 2018*). Third, a limited selection of PrCa cell lines is available because all are derived from metastases or healthy donors, thus limiting our ability to investigate MEIS function throughout tumor initiation and progression and requiring sub-optimal lentiviral over-expression approaches using cell lines from advanced metastases. Fourth, previous analyses of MEIS$^{high}$ vs. MEIS$^{low}$ prostate tumors and metastases did not prioritize proteoglycans or DCN. This could be accounted for since the RNA-Seq data were obtained from heterogeneous tumor tissues which likely harbored contaminating DCN-expressing stromal cells (*Bhanvadia et al., 2018*). Thus, while our results are supportive of the model identified in cell lines, they would likely benefit from further validation using additional models for PrCa progression as well as larger clinical datasets with annotation.

## Conclusion

Within a normal PrECs, MEIS1 complexes with HOXB13 to maintain expression of proteoglycans such as DCN, LUM, and TGFBR3 and repress growth factor and migration/invasion signaling through RTKs (*Figure 7C*). As a cell transforms to a malignant state, *MEIS1*/2 are epigenetically silenced in more aggressive prostate tumors (*Bhanvadia et al., 2018*) and expression of tumor-suppressive proteoglycans is suppressed, leading to decreased regulation of oncogenic signaling through pathways such as TGFβ, EGFR, cMYC, WNT, and IGF1R (*Figure 7C*). While this is likely not the only mechanism of MEIS-mediated tumor suppression, it does appear to have clinical significance. However, validation in larger clinical datasets is needed. Loss of MEIS1/2 expression also opens the possibility of new, non-canonical cofactors interacting with HOXB13 and further driving oncogenic signaling. An exciting possibility in this regard is the documented interaction between AR and HOXB13 that arises in malignant cells, and whether pharmacologic restoration of MEIS1 expression blocks oncogenic AR–HOXB13 interaction and thus impedes metastasis and castration-resistance. Further, a mechanism for increased risk and aggressiveness of PrCa observed with *HOXB13*$^{G84E}$ and *HOXB13*$^{G135E}$ mutations remains undefined. However, their locations in the MEIS-interacting domains of HOXB13 point toward changes to MEIS–HOXB13 complexes and/or transcriptional regulation as leading to decreased tumor suppression and enabling malignant transformation.

## Materials and methods

Key Resources Table is included as *Supplementary file 7*.

## Contact for reagent and resource sharing

Further information and requests for resources and reagents should be directed to and will be fulfilled by the Lead Contact and Project PI, Donald J. Vander Griend (dvanderg@uic.edu).

## Experimental model and subject details

### Mice

All mice used in this study were 4–6 week-old, male athymic nude mice (Harlan). All animal studies were carried out in strict accordance with the recommendations in the Guide for the Care and Use of Laboratory Animals of the National Institutes of Health. The protocol was approved by the

University of Chicago Institutional Animal Care and Use Committee (IACUC) (protocol #72231) as well as by the University of Illinois at Chicago IACUC (protocol #18–100).

## Cell lines

All cultures were routinely screened for the absence of mycoplasma contamination using the ATCC Universal Mycoplasma Detection Kit (#ATCC 30–1012K). Cell authentication of all lines was confirmed via the University of Arizona Genetics Core Facility (https://uagc.arl.arizona.edu/). Dr. John Issacs at The Johns Hopkins University generously provided the CWR22Rv1and LAPC4 cell lines, which were previously characterized (*van Bokhoven et al., 2003*). CWR22Rv1 cells were maintained in Roswell Park Memorial Institute 1640 (RPMI 1640) medium supplemented with 10% fetal bovine serum (FBS) and 1% penicillin/streptomycin. LAPC4 cells were maintained in Iscove's modified Eagle medium with 10% FBS, 1% penicillin/streptomycin, and 1 nM R1881 as required for survival (*Kregel et al., 2016*). HEK293T cells were maintained in Dulbecco's Modified Eagle Medium (DMEM) containing 5% FBS.

Primary epithelial cell shot-term cultures (PrECs) were established from fresh human prostate tissue obtained from surgical specimens as described previously (*Chen et al., 2012*; *Vander Griend et al., 2008*). Tissues were acquired under an expedited protocol approved by the University of Chicago Institutional Review Board (IRB #10–381-A). The University of Chicago Anatomic Pathology laboratory processed tissue samples. Patients' consents were waived because tissues were de-identified. Biopsy punches (4 mm) of non-tumor tissue were taken from prostate tissue removed during radical prostatectomies. Half of the punch was fixed and analyzed by a pathologist to confirm that lack of tumor. Dissociation of the remainder of the punch and subsequent outgrowth of cell cultures was performed as described previously (*Chen et al., 2012*; *Vander Griend et al., 2008*). PrEC cultures were grown in Keratinocyte Serum-Free Defined media supplemented with growth factors (#17005042, Thermo Fisher Scientific). For our experiments, all cultures were analyzed on or before their fourth passage.

## Method details

### In vivo tumor growth and metastasis assays

All surgery was performed under Ketamine/Xylazine anesthesia, and all efforts were made to minimize suffering. To measure tumor growth in a uniform androgen environment, host mice were surgically castrated at least one week before cell inoculation and simultaneously implanted with a 1.4 cm testosterone pellet subcutaneous. Mice were allowed to recover and testosterone levels to equilibrate for 7 days before tumor injections. In vivo tumor formation of derived lines from CWR22Rv1 cells were conducted via a sub-cutaneous inoculation of 250,000 cells into the flanks of 4–6 week-old, male athymic nude mice using a 75% Matrigel (Corning, Bedford, MA) and 25% HBSS solution (#14170–112, Thermo Fisher Scientific). Tumor length ($l$), width ($w$), and height ($h$) were measured using digital calipers and tumor volumes were determined using the formula $V=(pi/6) * l * w * h$ (*Tomayko and Reynolds, 1989*). Mice were euthanized when tumor burden exceeded 1,500 mm$^3$.

Intracardiac injections were performed in castrated 4–6 week old athymic nude mice (Envigo, previously Harlan) that were castrated two weeks prior to injection. 250,000 CWR22Rv1 cells in exponential growth phase, were suspended in 100 μL of PBS (Invitrogen), and injected into the left ventricle of the mice with a 28-gauge syringe. Metastatic colonization was visualized via Optical Imaging at least once a week post-injection until endpoint was met. Animals were imaged 10 min after IP injection of a 20 mg/kg bolus of D-Luciferin (#LUCNA-2G, GoldBio), once per week following intracardiac injection. Optical Imaging was performed on a PerkinElmer IVIS Spectrum with Living Image 4.4 Software at the University of Illinois at Chicago. Animals were euthanized at 20% wt loss from time of injection. Animals were dissected and organs harboring metastases were removed and fixed in formalin for 48 hr at room temp before being transferred into 70% ethanol. Tissues were then embedded in paraffin, sectioned and Hematoxylin and Eosin (H and E) stained by the Histology and Tissue Imaging Core at University of Illinois at Chicago. H and E stained sections were then imaged at 20x brightfield on an Aperio Slide Scanner.

## Lentiviral production and transduction

To derive cell lines overexpressing lentiviral constructs of MEIS1 or MEIS2, all lentiviral vectors used in this study contain the gene-of-interest in the pReceiver-LV105 backbone (GeneCopoeia). High titer lentivirus was made by separately co-transfecting the LV105 constructs with ViraPower Lentiviral packaging mix (#K497500, Thermo Fisher Scientific) in HEK-293T cells using Lipofectamine 2000 (#11668019, Thermo Fisher Scientific) according to manufacturer's instructions. After 48 and 72 hr, media containing the lentivirus was collected, spun down, and filtered using a 0.45 mm filter and used to infect target CWR-22rv1 or LAPC-4 cells with 5 mg/mL polybrene for 48 hr. Complete media were then replaced followed by selection and maintenance with puromycin (1 mg/mL, Invitrogen). Confirmation of MEIS1 or MEIS2 expression was confirmed using both qRT-PCR and Western blotting (anti-MEIS1 #ab19867, abcam); anti-MEIS2 (#TA337288, OriGene).

## CRISPR/Cas9-mediated Deletion of HOXB13

To achieve CRISPR generated knockout of HOXB13, parental CWR22rv1 and LAPC4 cells were seeded at $1 \times 10^6$ cells in a 10 cm dish. Cells were co-transfected with a 1:1 ratio of pT2-EF1a-Cas9-P2A-puro and pCMV(CAT)T7-SB100 (#34879, Addgene) using Lipofectamine 2000 following manufacturer guidelines. After 48 hr, cells with EF1a-Cas9-P2A-puro integrated into the genome by the SB100 transposase were selected for and maintained with puromycin (1 mg/mL, Invitrogen). Following 1 week of puromycin selection, Cas9 expression was confirmed by western blot (#14697, Cell Signaling Technologies). After confirmation of constitutive Cas9 expression, a custom crRNA (IDT) targeting the N-terminus of HOXB13 was selected using CHOPCHOP software (HOXB13 crRNA: 5'-TTGACAGCAGGCATCAGCGT-3') and was annealed with the tracrRNA-ATTO 550 (#1075927, IDT) according to manufacturer guidelines. A final concentration of 10 nM of the crRNA-tracrRNA duplex was then transfected into CWR22Rv1-Cas9 or LAPC4-Cas9 cells using siLentFect Lipid Reagent for RNAi (#1703360, Bio-Rad) according to manufacturer guidelines. Successful knockout of HOXB13 was confirmed by western blot (anti-HOXB13(F-9), #sc-28333, SCBT). Limiting dilution was then performed to establish 6 clonal knockout lines, each verified by western blot. The 6 knockout clones were then re-combined into one knockout pool deemed CWR22rv1- or LAPC4-HOXB13$^{ko}$. The HOXB13$^{ko}$ lines were then infected with LV-MEIS1 as described above to generate the HOXB13$^{ko}$-LV-MEIS1 lines for both CWR22Rv1 and LAPC4.

## Western blotting

Whole-cell lysates of 100,000 or more cells were used. Cells were rinsed with cold PBS and scraped into protein lysis buffer (20 mM Tris, pH 7.5; 150 mM NaCl; 1 mM EDTA; 1 mM EGTA; 2.5 mM sodium pyrophosphate; 1 mM sodium glycerophosphate; 1 mM sodium orthovanadate; 1% Triton-X 100) supplemented with cOmplete Mini Protease Inhibitor Cocktail (#11873580001, Sigma-Aldrich), sonicated on ice for 10 s at 30% amplitude. The Pierce BCA Protein Assay Kit (#23227, Thermo Fisher Scientific) was used to determine protein concentration according to manufacturer directions. Forty micrograms of protein per lane was resuspended in 5x Laemmli Sample Buffer supplemented with 10% b-mercaptoethanol and boiled at 95°C for 5 min. Samples were loaded on a 10% SDS-polyacrylamide gel, and SDS-PAGE was carried out in Tris/Glycine/SDS Buffer for 1 hr at 120 V. Protein was transferred to nitrocellulose membranes with Tris/Glycine Buffer containing 20% methanol for 1.5 hr at 4°C with 400 mA. Nitrocellulose membranes were blocked with TBS+5% nonfat milk for 1 hr at room temperature. Primary antibodies were applied in TBST+5% nonfat milk overnight at 4°C at the noted dilutions. (Anti-β-actin 1:10000; Anti-MEIS1 1:1500; Anti-MEIS2 1:1000; Anti-HOXB13 (F-9) 1:100; Anti-DCN 1:500; Anti-TGFBR3 1:200; Anti-LUM 1:300; Anti-ITGB1 1:1000; Anti-EGFR 1:1000; Anti-EGFR-phos 1:500; Anti-c-MYC 1:1000; Anti c-MYC-pT58 1:500; Anti-SMAD2/3 1:500; Anti-SMAD2-phos 1:200; Anti-p21 1:500). Membranes were washed in TBST 3 times for 5 min. Secondary antibodies were applied in TBST+5% nonfat milk at 1:10,000 for 1 hr at room temperature. Membranes were washed in TBST 3 times for 5 min. Blots were scanned with an Odyssey imaging system (LI-COR Biosciences) and analyzed with LI-COR Image Studio software.

## Co-Immunoprecipitation

Whole cell lysates from PrECs, CWR22Rv1-Cas9, CWR22Rv1-LV-MEIS1, LAPC4-Cas9 and LAPC4-LV-MEIS1 cells were prepared in protein lysis buffer (20 mM Tris, pH 7.5; 150 mM NaCl; 1 mM EDTA; 1

mM EGTA; 2.5 mM sodium pyrophosphate; 1 mM sodium glycerophosphate; 1 mM sodium orthovanadate; 1% Triton-X 100) supplemented with cOmplete Mini Protease Inhibitor Cocktail (#11873580001, Sigma-Aldrich). To pull down MEIS1, 1000 µg of precleared cell lysates were incubated with 2 µg of Anti-HOXB13 (EPR17371; Cat #ab201682, Abcam) antibody overnight at 4℃. The lysates were then incubated with immobilized Protein A/G Agarose beads (#20421, Thermo Scientific, USA) overnight at 4℃. Finally, the beads were washed three time with protein lysis buffer and centrifuged at 600 g for 5 min. Co-immunoprecipitated proteins were then eluted from the beads by adding 40 µL of RIPA buffer to 5 × SDS PAGE sample buffer and heating at 95℃ for 5 min. Samples were further size-fractionated on 10% SDS-polyacrylamide gels. The resolved gels were electro-transferred onto nitrocellulose membranes and probed for Anti-MEIS1 (1:1000, Cat #0T12A3, Origene) and Anti-HOXB13 (1:1000) with respective secondary antibodies (1:10000). The membranes were scanned on Odyssey imaging system (LI-COR Biosciences) and analyzed with LI-COR Image Studio software.

## Total RNA isolation

Cultured cells (300,000) were lysed in Buffer RLT containing 1% 2-mercaptoethanol and homogenized with a 28-gauge needle syringe. Subsequent total RNA extraction and DNase treatment of samples was performed using the RNeasy Mini Kit (#74106, Qiagen) and the RNase-Free DNase Set (#79254, Qiagen) according to manufacturer directions. Purified RNA was quantified on a Synergy LX Multi-mode Reader with a Take 3 plate (BioTek) and quality tested for RIN score >7 using an Agilent Bioanalyzer 2100 (Agilent Technologies).

## Reverse transcription and qRT-PCR

Reverse transcription of RNA to cDNA was carried out using qScript cDNA SuperMix (#95048–100, QuantaBio) according to manufacturer protocol starting with 1 µg total RNA per reaction. qRT-PCR was done on a Roche LightCycler 96 using Power SYBR Green Master Mix (#4368702, Life Technologies). Reactions were performed in 20 µL volumes (10 µL 2x Power SYBR Green Master Mix; 1 µL of 10 µM forward primer; 1 µL of 10 µM reverse primer; 50 ng cDNA; 7 µL nuclease free water). Relative expression of cDNA was normalized by the ΔΔCt method using *RPL13A* as a housekeeping gene.

## RNA sequencing

Total RNA was purified as described above. Illumina sequencing libraries were prepared with the KAPA mRNA-Seq Kit (#KK8420, KAPA Biosystems) following manufacturer's protocol starting from 2 µg total RNA and aiming for fragment size of 100–200 bp before addition of adapters. All libraries received 9 cycles of amplification. Quality of the enriched libraries was validated using the 2100 TapeStation System aiming for an average final fragment size of approximately 300–350 bp. The libraries were quantified using the Library Quantification Kit – Illumina/Universal Kit (#KK4824, KAPA Biosystems) and evenly pooled together by molarity for 12 libraries per lane. Sequencing was performed by the Functional Genomics Core Facility at the University of Chicago on a HiSeq 4000 Sequencing System (Illumina) with 50 bp single-end reads.

## RNA sequencing analysis

The quality of raw reads was accessed by FastQC (v0.11.4). Adapter sequences and low-quality reads were trimmed using Trimmomatic-0.38. All reads were pseudo-aligned to the human transcriptome built from ENSEMBL Human GRCh38.p12 cDNA and ncRNA using kallisto (v0.43.1) with default settings for single-end reads with fragment length of 180 and standard deviation of 20. Estimated transcript counts were summarized to the gene level using tximport (v1.8.0) and filtering of lowly-expressed genes (<10 counts in half the samples of one group), library normalization, and differential expression analysis was carried out in edgeR (v3.22.2) using the glmTreat method with a log fold-change threshold of $\log_2(1.5)$ and FDR < 0.05. Further biological insights were gained by performing Gene Set Enrichment Analysis (GSEA 3.0) from Broad Institute on MSigDB collections for Oncogenic Signaling and for Gene Ontology: Biological Function gene sets. Pathway analyses were performed using Enrichr online tool.

## Chromatin isolation and immunoprecipitation

Chromatin isolation of 25 million cells per cell line and chromatin immunoprecipitation (ChIP) was performed with the iDeal ChIP-seq Kit for Transcription Factors (#C01010055, Diagenode) according to manufacturer's guidelines. Chromatin was sheared to a size of 200–500 bp using a Bioruptor Pico (Diagenode) and shearing efficiency was verified by agarose gel electrophoresis. The ChIP was performed with 6 µg of ChIP-grade antibody against MEIS1 (#ab19867, abcam) or IgG control antibody (#3900, Cell Signaling Technology). Anti-MEIS1 ChIP was performed in biological triplicate in CWR22Rv1 LV-MEIS1 cells and in duplicate from CWR22Rv1 HOXB13ko-LV-MEIS1 cells. Verification of immunoprecipitation of target protein and chromatin was verified by western blot. Final immuno-precipitated chromatin concentration was determined with a Qubit dsDNA HS Assay Kit (#Q32851, Thermo Fisher Scientific).

## ChIP sequencing

ChIP sequencing libraries were generated using the Low Throughput Library Prep Kit (#KK8230, Kapa Biosystems) according to manufacturer protocol. 5 ng of either cell line specific Input chroma-tin or MEIS1-ChIP'd chromatin were used to begin the protocol. All libraries underwent 13 cycles of amplification. Quality of the enriched libraries was validated using the 2100 TapeStation System. The libraries were quantified using the Library Quantification Kit – Illumina/Universal Kit and evenly pooled together by molarity for six libraries per lane. Sequencing was performed by the Functional Genomics Core Facility at the University of Chicago on a HiSeq 4000 Sequencing System (Illumina) with 50 bp single-end reads.

## ChIP sequencing analysis

The quality of raw reads was accessed by FastQC (v0.11.4). Adapter sequences and low-quality reads were trimmed using Trimmomatic-0.38. Duplicate reads were marked and removed using Pic-ard Tools (v2.18.10). Processed reads were then aligned to human reference genome hg38 (ENSEMBL release 93, GRCh38.p12) using Bowtie2 with default settings. Samtools was then used to change. sam files to. bam files, sort, and index the. bam files. Peaks were called against sequenced input chromatin using MACS2 and pooling immunoprecipitated samples for either LV-MEIS1 or HOXB13ko-LV-MEIS1 cells. Called peaks were then annotated to hg38 using the annotatePeaks.pl command in HOMER (v4.9.1). DeepTools (v3.2.0) was also used for generation of enrichment heat-maps and bigwig files. SpaMo analysis was performed with MEME Suite online tool, and motif simi-larity assessment was performed using STAMP online tool.

## ChIP-qPCR

Input chromatin or MEIS1-ChIP chromatin from CWR22Rv1-LV-MEIS1 cells and CWR22Rv1-HOXB13ko-LV-MEIS1 cells were used with 500 pg DNA per reaction. qPCR was done on a LightCy-cler 96 using Power SYBR Green Master Mix (Cat #4368702, Life Technologies). Reactions were per-formed in 20 µL volumes (10 µL 2x Power SYBR Green Master Mix; 0.5 µL of 10 µM forward primer; 0.5 µL of 10 µM reverse primer; 500 pg of Input chromatin or MEIS1-ChIP chromatin; 8 µL nuclease free water). Relative binding of MEIS1 to DCN, LUM, and TGFβR3 was normalized to expression lev-els of Negative and Positive Contorl ACTB-1 ChIP Primer sets (Active Motif; Carlsbad, CA).

## Proximity Ligation Assay (PLA)

Cells were seeded at $5 \times 10^5$ cells per well in an 8-well glass chamber slide and incubated at 37°C and 5% CO2 for 24 hr. Next, cells were fixed with 4% paraformaldehyde in PBS for 20 min on ice, with gentle shaking. Cells were then quenched with 50 mM ammonium chloride in PBS for 10 min. Cells were then washed 3 times for 5 min each at room temperature with PBS. Cells were then quickly rinsed with MilliQ ddH2O to remove any salts. Plastic chambers were then removed from the slides and reaction areas were delimited with a grease pen on the boarder of each well. Next, cells were permeabilized with 0.3% TritonX-100 in PBS for 20 min at room temperature before being washed 3 times for 5 min each with PBS. At this point, the manufacturer protocol for Duolink In Situ Red Starter Kit Mouse/Rabbit (#DUO92101, Sigma-Aldrich) was followed as directed, starting at adding 1 drop of Duolink Blocking Solution to each well of the 8-well slide and incubating for 30 min at 37°C. Primary antibodies against proteins-of-interest used were anti-HOXB13(F-9) (1:100, #sc-

28333, Santa Cruz Biotechnology) and anti-MEIS1 (1:1000, #ab19867, Abcam). Secondary-antibody-only controls as well as controls individually replacing each of the primary antibodies with the species matched IgG (# 3900, Cell signaling technologies; # 5415, Cell signaling technologies) were used to ensure signals were not background. Imaging of slides were done on the Keyence BZX-800 with a 60x-oil objective. All images were taken as 10-micron thick z-stacks with a 0.4-micron step size and max projections of each stack were used for image analysis. Image analysis was done using Fiji (*Schindelin et al., 2012*) to count the number of foci observed per individual nucleus. Foci that did not overlap with the nuclear DAPI signal were considered to be background and ignored.

## MEIS2 isoform identification

The presence or absence of each MEIS2 isoform was determined by sequencing individual transcripts via TOPO-TA cloning. MEIS2 transcripts were amplified using pan-MEIS2 PCR primers and fragments inserted into a pCR4-TOPO-TA cloning vector (Invitrogen). Each bacterial colony thus represented a single isoform. 100 colonies per cell line were sequenced using T3 and T7 sequencing primers and analyzed using standard Sanger sequencing. A 5% cutoff was considered to indicate the presence of a particular isoform, whereas below that threshold the presence of the isoform is not definitive. MEIS2E specific primers were designed and used to further confirm the absence of MEIS2E in PrECs.

## Cell proliferation assays

Three different measurement techniques were used to assay cell proliferation in this paper. The first method used was physical counting of cells with the assistance of a Cellometer Auto T4 Bright Field Cell Counter. Cells were plated at 250 k cells per dish into three 60 mm dishes per line and per time-point (i.e. for a five-day experiment with cells counted every 24 hr, one cell line would start with fifteen 60 mm dishes – three dishes per timepoint). Every 24 hr three dishes per cell line were trypsinized, re-suspended, sent through a filter cap to achieve single cell suspension, mixed with trypan blue to avoid counting dead cells, and counted on Cellometer Auto T4 Bright Field Cell Counter. Counts were performed in triplicate per dish, per timepoint and averaged.

The second and third methods for assessing proliferation both used the CyQUANT Direct Cell Proliferation Assay (#C35011, Thermo Fisher Scientific) and either 1) measured relative fluorescence in a well using a SpectraMax i3x Multi-Mode Microplate Reader (#i3x, Molecular Devices) at 480/535 nm excitation/emission wavelengths; or 2) imaging green fluorescence in entire wells at 10x, stitching images, and performing automated cell counting on the Keyence BZX-800 all-in-one fluorescence microscope. The switch to imaging rather than relative fluorescence was made due to lack of availability of the SpectraMax i3x plate reader with bottom-read capabilities at later dates. The relative fluorescence measurement method is dependent on a bottom-read capable plate reader. For both methodologies using the CyQuant direct cell proliferation kit, 1,500 cells per well were plated in black-wall, clear-bottom 96-well plates (#353219, Corning) in complete growth medium and incubated at 37C with 5% CO2. Plates were then read according to the manufacturer protocol with either the SpectraMax i3x or Keyence BZ X-800 every 24 hr post seeding of the cells.

## Cell cycle assay

Cell cycle was determined on a Cellometer Spectrum by way of propidium iodide (PI) fluorescence intensity according to manufacturer protocol (#CSK-0112, Nexcelom Bioscience) (*Chan et al., 2011*). Briefly, cells were plated 48 hr before assay and grown to ~80% confluence. Cells were then trypsinized, filtered to a single cell solution, fixed with ice cold ethanol, treated with PI and RNAse A, washed, and loaded into Cellometer spectrum for imaging and counting. Cell count, size, and PI intensity data were then exported to FCS Express six cytometry software where proper gating was determined for G0/G1, S phase, and G2/M cell cycle phases based on cell size, count, and PI intensity. Each cell line was performed in triplicate.

## TUNEL assay

Cell death due to MEIS1 or MEIS2 exogenous expression was determined by Click-iT TUNEL Alexa Fluor 647 Imaging Assay, for microscopy and HCS kit according to manufacturer protocol (# C10247, Thermo Fisher Scientific). Briefly, Control, MEIS1, or MEIS2 expressing cells were plated in

clear-bottom, black-wall, 96-well plates at 2,500 cells per well and allowed to grow for 48 hr in normal growth media. An extra group of Control expressing cells was plated to be used as a positive control for the assay by treating this group with DNAse 1. Cells were fixed, permeabilized and treated with TdT reaction buffer followed by the Click-it reaction buffer. AlexaFluor 647 was used as a secondary antibody to visualize successful TdT reactions marking dead cells, and DNA was counterstained with Hoechst to visualize nuclei. Complete wells were imaged on the Keyence BZ X-800 microscope with both a DAPI and a far red filter at 4x. Images were stitched together for each well and double positive nuclei (Hoechst and AF647) were counted manually with the aid of Fiji software.

## AO/PI Cell Viability Assay

Cell viability over time in LAPC4-Control, -LV-MEIS1, -HOXB13ko, and -HOXB13ko-LV-MEIS1 was assessed using the ViaStain AOPI Staining Solution (#CS2-0106, Nexcelom) and Cellometer spectrum according to manufacturer guidelines. Briefly, cells were plated at $1.0 \times 10^5$ in 12 well dishes in complete media and allowed to incubate at 37C and 5% $CO_2$ until specified timepoints at 24, 48, and 96 hr. At each time point, the conditioned media, PBS wash, and trypsinized cells were all collected and spun down at 300 x g for four mins. Cells were resuspended in 1 mL of media and mixed 1:1 with Viastain AOPI Staining solution. Live and dead cells were then counted on a Cellometer Spectrum with with the manufacturer defined program for AO/PI Viability that uses Red and green fluorescence as well as brightfield. Four counts from each cell line were performed at each timepoint.

## Cell migration assay

Corning Transwell polycarbonate membrane cell culture inserts, 8 µm pore size (#CLS3422, Corning) were used to measure cell migration. Cells were serum starved for 24 hr prior to seeding. Cells were seeded at $1.5 \times 10^5$ per insert to the top half of each insert in serum free media. The attracting media on the underside of each insert was complete growth media with 10% FBS. All media (top and bottom) also contained 3 µM Aphidicolin (#14007, Cayman Chemical Company) to inhibit proliferation in order to decrease the confounding effect of differences seen in proliferation rates seen between Control and MEIS1 or MEIS2 expressing cells. Aphidicolin has been shown to have no impact migration (*Müller et al., 2002*). After 48 hr, non-migrated cells were removed from the top of the insert using a cotton swab. Transwell membranes were then fixed and stained with Crystal Violet staining solution (50 mg Crystal Violet; 2.7 mL 37% formaldehyde; 1 mL methanol; 96.3 mL 1x PBS) for 20 mins at room temp. After staining, excess stain was washed away by dunking inserts into 6 consecutive, 100 mL aliquots of ddH$_2$O. Inserts were then allowed to air dry for 30mins before membranes were cut out using a razor blade and mounted on imaging slides with CytoSeal60 (#8310–4, Thermo Fisher Scientific.). Each complete insert was imaged at 10x brightfield on the Keyence BZ X-800 microscope and the Hybrid cell counter function of Keyence software was used to automate the counting of migrated cells stained by crystal violet.

## siRNA knockdown of decorin

For siRNA-mediated knockdown of DCN, we utilized a commercially available pre-validated pool of siRNAs targeting DCN (# L-021491-00-0005, Dharmacon), as well as a pool of non-targeting control siRNAs (# D-001810-10-05, Dharmacon). Cells were plated 24 hr prior to transfection and allowed to reach 50–60% confluence. siRNA (10 nM per plate) and siLentFect lipid transfection reagent (# 1703360, BioRad) were prepared according to manufacturer instructions in OptiMEM transfection media. Cells were exposed to transfection media containing siRNAs and siLentFect reagent for 18 hr before switching to normal growth media. DCN knockdown was confirmed at 72 hr post transfection by RT-PCR and western blot. Further experiments were performed as described above.

## Quantification and statistical analysis

Statistical analyses are as noted in each figure legend and were performed using GraphPad Prism 7 or R. For comparison of two groups, p values were calculated with a one-sided unpaired Student's *t*-test or Welch's two-tailed t-test for samples with unequal variance. For comparison of overall survival of mice in intracardiac study, the survdiff functionality of the R package, survival, was used. Survdiff makes use of log-rank and Chi square tests to determine significance between groups. Adjusted

p-values or FDR for all sequencing data was done using Benjamini-Hochberg method. All error bars represent standard error of the mean (SEM). Asterisks (*) always indicate significant differences as *=$p < 0.05$; ns = not significant, and n = number of replicates, unless otherwise specified.

## Acknowledgements

We wish to acknowledge Vander Griend and Szmulewitz lab members for their input. We wish to acknowledge Dr. Brendan Looyenga at Calvin College for his gift of vectors used for generation of knockout cell lines using CRISPR. We wish to acknowledge Drs. Lev Becker, Megan McNerney, and Russell Szmulewitz for their input on the direction of the project. We wish to acknowledge support of the University of Illinois at Chicago Department of Pathology led by Dr. Fred Behm, as well as the University of Illinois at Chicago Research Histology and Tissue Imaging Core led by Dr. Peter Gann. We wish to acknowledge the University of Chicago Genomics Facility led by Dr. Pieter Faber. We would also like to acknowledge support of the University of Chicago Committee on Cancer Biology led by Dr. Kay Macleod. Finally, we wish to acknowledge support of Drs. Alan Diamond, Larisa Nonn, and Gail Prins at the University of Illinois at Chicago Departments of Pathology and Urology.

## Additional information

### Funding

| Funder | Grant reference number | Author |
| --- | --- | --- |
| U.S. Department of Defense | PC130587 | Donald J Vander Griend |
| U.S. Department of Defense | PC180414 | Donald J Vander Griend |
| National Institutes of Health | P50 CA180995 | Donald J Vander Griend |
| National Institutes of Health | T32 CA009594 | Calvin VanOpstall Hannah Brechka Donald J Vander Griend |
| National Institutes of Health | F31CA232651 | Calvin VanOpstall |
| National Institutes of Health | P30CA014599 | Donald J Vander Griend |

The funders had no role in study design, data collection and interpretation, or the decision to submit the work for publication.

### Author contributions

Calvin VanOpstall, Conceptualization, Data curation, Formal analysis, Investigation, Methodology, Writing - original draft, Writing - review and editing; Srikanth Perike, Conceptualization, Formal analysis, Validation, Investigation, Writing - review and editing; Hannah Brechka, Conceptualization, Formal analysis, Investigation, Methodology, Writing - original draft; Marc Gillard, Formal analysis, Validation, Investigation, Methodology; Sophia Lamperis, Data curation, Formal analysis, Validation, Investigation; Baizhen Zhu, Data curation, Investigation; Ryan Brown, Data curation, Formal analysis, Investigation; Raj Bhanvadia, Conceptualization, Formal analysis, Investigation, Methodology; Donald J Vander Griend, Conceptualization, Data curation, Formal analysis, Supervision, Funding acquisition, Investigation, Methodology, Writing - original draft, Project administration, Writing - review and editing

### Author ORCIDs

Donald J Vander Griend https://orcid.org/0000-0003-4421-5698

### Ethics

Animal experimentation: All animal studies were carried out in strict accordance with the recommendations in the Guide for the Care and Use of Laboratory Animals of the National Institutes of Health. The protocol was approved by the University of Chicago Institutional Animal Care and Use Committee (IACUC) (protocol #72231) as well as by the University of Illinois at Chicago IACUC (protocol #18-100).

Decision letter and Author response
Decision letter https://doi.org/10.7554/eLife.53600.sa1
Author response https://doi.org/10.7554/eLife.53600.sa2

# Additional files

## Supplementary files

• Supplementary file 1. MEIS1 ChIP-seq peaks and annotation in CWR22Rv1 LV-MEIS1, Related to *Figure 5*. Peaks identified by MACS2 in LV-MEIS1 ChIP-seq samples vs. input DNA were annotated to the nearest transcription start site (TSS) using HOMER.

• Supplementary file 2. MEIS1 ChIP-seq peaks and annotation in CWR22Rv1 HOXB13ko-LV-MEIS1, Related to *Figure 5*. Peaks identified by MACS2 in HOXB13ko-LV-MEIS1ChIP-seq samples vs. input DNA were annotated to the nearest transcription start site (TSS) using HOMER.

• Supplementary file 3. Counts per million (CPM) expression of genes in RNA-seq from CWR22Rv1 cell line derivatives, Related to *Figure 5*. Adapters and low-quality reads were trimmed before aligning sequences to hg38 transcriptome (ENSEMBL) using kallisto. Transcript counts were then summarized to the gene level with tximport and further analyzed with EdgeR to remove genes with low counts and normalize to library sizes.

• Supplementary file 4. All significantly differentially expressed genes between CWR22Rv1-LV-MEIS1 and control cells, Related to *Figure 5*. TREAT and GLM methodologies in edgeR were used to determine significantly differentially expressed genes (fold-change >±1.5, FDR < 0.05) in LV-MEIS1 vs. control cells.

• Supplementary file 5. DEGs that are direct targets of MEIS1 only when HOXB13 is present, Related to *Figure 5*. Overlap of DEGs between LV-MEIS1 and control cells with ChIP-seq targets from both HOXB13ko and HOXB13ko-LV-MEIS1 cells identified 157 DEGs that are targets of MEIS1 only when HOXB13 is present.

• Supplementary file 6. GSEA for Gene Ontology: Biological Processes on RNA-seq from LV-MEIS1 and control cells, Related to *Figure 5*. The top 20 enriched gene sets from CWR22Rv1 RNA-seq of GSEA on the Gene Ontology: Biological Processes collection from MSigDB.

• Supplementary file 7. Key resources table.

• Transparent reporting form

## Data availability

RNA-seq and ChIP-seq raw and analyzed data have been deposited at the Gene Expression Omnibus and Sequence Read Archive under the accession number GSE132717.

The following dataset was generated:

| Author(s) | Year | Dataset title | Dataset URL | Database and Identifier |
|---|---|---|---|---|
| VanOpstall C, Vander Griend DJ | 2020 | MEIS-mediated suppression of human prostate cancer growth and metastasis through HOXB13-dependent regulation of proteoglycans | https://www.ncbi.nlm.nih.gov/geo/query/acc.cgi?acc=GSE132717 | NCBI Gene Expression Omnibus, GSE132717 |

The following previously published datasets were used:

| Author(s) | Year | Dataset title | Dataset URL | Database and Identifier |
|---|---|---|---|---|
| Robinson | 2015 | Integrative clinical genomics of advanced prostate cancer | https://www.ncbi.nlm.nih.gov/projects/gap/cgi-bin/study.cgi?study_id=phs000915.v1.p1 | NCBI dbGaP, phs000915.v1.p1 |
| Pflueger | 2011 | Discovery of non-ETS gene fusions | https://www.ncbi.nlm. | NCBI dbGaP, phs000 |

| in human prostate cancer using next-generation RNA sequencing | nih.gov/projects/gap/cgi-bin/study.cgi?study_id=phs000310.v1.p1 | 310.v1.p1 |
|---|---|---|

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
