## [Decision Letter]

**Acceptance summary:**

The authors convincingly show that tumor suppressor MEIS1 is functionally dependent on the prostate lineage specific transcription factor HOXB13, and provide a direct mechanistic explanation of MEIS1-driven tumor suppression in prostate cancer.

**Decision letter after peer review:**

Thank you for submitting your article "MEIS-mediated suppression of prostate cancer growth and metastasis through HOXB13-dependent regulation of proteoglycans" for consideration by *eLife*. Your article has been reviewed by three peer reviewers, including Wilbert Zwart as the Reviewing Editor and Reviewer #1, and the evaluation has been overseen by Richard White as the Senior Editor. The following individuals involved in review of your submission has agreed to reveal their identity: Ralph Buttyan (Reviewer #3).

The reviewers have discussed the reviews with one another and the Reviewing Editor has drafted this decision to help you prepare a revised submission.

Previous studies from the authors and others have shown that MEIS1 and 2 interact with and act as cofactors for HOXB13 in prostate cancer (PC), that their expression is downregulated with PC progression, and that they have tumor suppressor activity in PC. However, the precise mechanisms through which MEIS1 or 2 act, and the extent to which their functions are mediated through effects on HOXB13, have not been established. This study further establishes that HOXB13 and MEIS1 interact in PC cells, that MEIS1 overexpression can suppress tumor growth/invasion, and indicates that these effects are dependent on HOXB13. The major novel data are related to the MEIS1 cistrome and transcriptome, which indicate that MEIS1 in conjunction with HOXB13 increases the expression of proteoglycans including DCN, which account at least in part for MEIS1 tumor suppressor functions. The physiological significance is supported by gene expression data showing a positive correlation between MEIS1 and proteoglycans including DCN, LUM, and TGFBR3 in clinical data sets.

Essential revisions:

1) The PLA assays of Figure 2 require independent confirmation using alternative technologies, such as ChIP-reChIP or co-IP. In addition, in the Results section describing these findings, it is stated that MEIS1 loss decreases HOXB13/MEIS1 interactions (which sounds quite logical). However, same goes for HOXB13 mutation, but this is not shown in the paper, nor referred to another paper describing this. These data should be added if the claim is made

2) In Figure 5, a number of direct MEIS1/HOXB13 target genes are being studied. How is direct regulation defined? Being enhancer-acting transcription factors, direct control of genes is not trivial to determine. In addition, please include genome browser snapshots for HOXB13 ChIP-seq (which could be public data) and MEIS1 ChIP-seq at the relevant gene loci of Figure 5F.

3) For the MEIS1 sites that are being lost upon HOXB13ko in Figure 5A, are those also the genomic locations of HOXB13 binding?

4) Figure 1 indicates that MEIS1 protein is virtually absent in a series of PC cell lines, which would clearly make studies of the endogenous protein problematic. However, a previous study from the authors found readily detectable levels of MEIS1/2 in LAPC4 cells and carried out MEIS1 and 2 RNAi studies (Bhanvadia et al., 2018). Moreover, a recent study also found substantial MEIS1 levels in LAPC4 cells, with lower (but readily detectable) levels in other PC cell lines (Johng et al., 2019). Therefore, the conclusion that these lines express undetectable levels of MEIS1 should be clarified.

5) In the functional studies in Figure 1, the levels of MEIS1 protein expressed in the LV-MEIS1 cells should be shown in comparison to endogenous levels. Moreover, although it is possible that the endogenous levels may be too low to be functionally significant, the authors should assess effects of depleting endogenous MEIS1. In this regard, their previous paper (Bhanvadia et al., 2018) did show gene expression alterations in response to MEIS1/2 siRNA.

6) The major concern is that the ChIP-seq data in Figure 5 is based only on overexpressed exogenous MEIS1. It would again be important to assess MEIS1 levels relative to endogenous levels, and ideally to validate the ChIP-seq results using endogenous MEIS1.

7) The RNA-seq data should also be confirmed by MEIS1 knockdown. Indeed, the authors previously reported RNA-seq data in LAPC4 cells after MEIS1/2 knockdown (Bhanvadia et al., 2018).

8) Their previous publication also identified gene sets that were enriched in MEIS high versus low tumors, and did not identify proteoglycans. This may be because the current analysis is based on putative direct MEIS1 regulated genes, but this should in any case be discussed.

---

## [Author Response]

Essential revisions:1) The PLA assays of Figure 2 require independent confirmation using alternative technologies, such as ChIP-reChIP or co-IP. In addition, in the Results section describing these findings, it is stated that MEIS1 loss decreases HOXB13/MEIS1 interactions (which sounds quite logical). However, same goes for HOXB13 mutation, but this is not shown in the paper, nor referred to another paper describing this. These data should be added if the claim is made

We thank the reviewer for this feedback. We have conducted additional Co-IP experiments demonstrating that, in CWR22Rv1 and LAPC4 cells, MEIS1 pulls down with HOXB13 protein, and when MEIS1 is over-expressed there is increased MEIS1 pulldown. This is documented as a new Figure 2D, revised Results section, and new Materials and methods. With regards to HOXB13 mutations, we modified the Results and Discussion to include greater attention to Jhong et al., where they investigated HOXB13 mutations and their interaction with MEIS1.

2) In Figure 5, a number of direct MEIS1/HOXB13 target genes are being studied. How is direct regulation defined? Being enhancer-acting transcription factors, direct control of genes is not trivial to determine. In addition, please include genome browser snapshots for HOXB13 ChIP-seq (which could be public data) and MEIS1 ChIP-seq at the relevant gene loci of Figure 5F.

This is a great point, and to address this we’ve included additional analyses and new experimental data. We revised the text to clearly define direct regulation as MEIS1 binding via ChIP, increased expression when MEIS1 is ectopically expressed, and diminished MEIS1 binding and expression when HOXB13 is deleted using CRISPR. To solidify this conclusion, we have added Genome Browser snapshots demonstrating MEIS1 binding at the DCN, LUM, and TGFBR3 loci when HOXB13 is present, and loss of binding when HOXB13 is deleted (new Figure 5F). Further, we conducted site-specific ChIP-qPCR on MEIS binding regions within the DCN, LUM, and TGFBR1 promoter (new Figure 5G). These data clearly show MEIS1 binding when HOXB13 is present, and significantly diminished MEIS1 binding when HOXB13 is deleted.

3) For the MEIS1 sites that are being lost upon HOXB13ko in Figure 5A, are those also the genomic locations of HOXB13 binding?

This is a great question, which we’ve attempted to address three ways. First, we show that HOXB13 binding sites were significantly enriched at 1bp distance from MEIS1 binding sites (Figure 5B, p=2.84e-12). Second, we attempted three HOXB13 ChIP-seq experiments as part of this manuscript, and in all three instance we were not satisfied with the quality of sequencing data to include in our analyses. Third, and we attempted to mine publicly available HOXB13 ChIP-seq datasets to determine whether HOXB13 is binding at the DCN, LUM, and TGFBR1 loci (see Author response image 1). In particular, the Pomerantz et al. dataset (Pomerantz et al., 2015) utilized VCAP and LNCaP cells, and other datasets utilized over-expressed HOXB13, both of which severely hindered our ability to robustly compare datasets to our MEIS1-expressing datasets. We continue to optimize our HOXB13 ChIP-seq approach, but feel these data will be more suitable for a future manuscript. These points have been included in a revised Discussion section.

**Author response image 1. sa2fig1:** This file contains the HOXB13 binding to DCN and LUM of every publicly available HOXB13 ChIP-seq experiment done in prostate tissue or cell lines. Every column is a different study or different treatment condition from the same study. The SRX###### number at the top of each column to be taken to more info about the study/experimental design for that HOXB13 ChIP run. The numbers for everything reflect the HOXB13 peak intensity (high number = more confident peak).

4) Figure 1 indicates that MEIS1 protein is virtually absent in a series of PC cell lines, which would clearly make studies of the endogenous protein problematic. However, a previous study from the authors found readily detectable levels of MEIS1/2 in LAPC4 cells and carried out MEIS1 and 2 RNAi studies (Bhanvadia et al., 2018). Moreover, a recent study also found substantial MEIS1 levels in LAPC4 cells, with lower (but readily detectable) levels in other PC cell lines (Johng et al., 2019). Therefore, the conclusion that these lines express undetectable levels of MEIS1 should be clarified.

We thank the reviewer for this feedback. We have re-done the Western blots and revised the Results section to clearly document MEIS1 and MEIS2 levels in prostate cancer cells. This includes a revised Figure 1B that includes a PrEC control lysate.

5) In the functional studies in Figure 1, the levels of MEIS1 protein expressed in the LV-MEIS1 cells should be shown in comparison to endogenous levels. Moreover, although it is possible that the endogenous levels may be too low to be functionally significant, the authors should assess effects of depleting endogenous MEIS1. In this regard, their previous paper (Bhanvadia et al., 2018) did show gene expression alterations in response to MEIS1/2 siRNA.

Thank you for pointing out this need for clarification. We attempted several western blots to clearly demonstrate differences in protein expression between parental, control, and LV-MEIS1-expressing cells. However, the contrast in protein expression between controls and ectopic expression was difficult to clearly illustrate for MEIS1 and MEIS2. We have, however, added new qPCR data demonstrating changes in mRNA levels in response to MEIS over-expression (revised Figure 1C). With regards to the shRNA-MEIS1/2 siRNA, our 2018 paper (Bhanvadia et al., 2018) demonstrated that dual MEIS1/MEIS2 knockdown was necessary to achieve a phenotypic change (increased tumor xenograft growth), which is why these functional studies primarily focused on ectopic expression of MEIS1 with HOXB13 modulation. These have been clarified in the Results and Discussion sections.

6) The major concern is that the ChIP-seq data in Figure 5 is based only on overexpressed exogenous MEIS1. It would again be important to assess MEIS1 levels relative to endogenous levels, and ideally to validate the ChIP-seq results using endogenous MEIS1.

We thank the reviewer for this suggestion. The levels of endogenous MEIS1 in CWR22Rv1 and LAPC4 cells is below a level of expression where we can reliably create ChIP-sequencing libraries with sufficient quality to sequence. This is primarily due to our necessary utilization of a ChIP-grade MEIS1 antibody, which demonstrates adequate specificity for MEIS1, albeit with below-average sensitivity. We are currently creating new anti-MEIS antibodies which may enable reliable MEIS1 ChIP from cell lines expressing low levels.

7) The RNA-seq data should also be confirmed by MEIS1 knockdown. Indeed, the authors previously reported RNA-seq data in LAPC4 cells after MEIS1/2 knockdown (Bhanvadia et al., 2018).

This is a great recommendation, and we have analyzed our LAPC4 cells with MEIS1/2 knockdown for changes in DCN protein levels. Indeed, knockdown of either MEIS1 or MEIS2 decreases DCN protein expression, and dual MEIS1/2 shRNA-knockdown rendered DCN protein levels nearly undetectable. This is included as a new Figure 5—figure supplement 2, and we have referenced and discussed this new data in the Results section.

8) Their previous publication also identified gene sets that were enriched in MEIS high versus low tumors, and did not identify proteoglycans. This may be because the current analysis is based on putative direct MEIS1 regulated genes, but this should in any case be discussed.

We appreciate this suggestion and have included the in the Discussion how our current data can be explained in the context of our previous publication. We believe that proteoglycans were not prioritized in our previous analyses because the datasets include minor stromal contamination, which are known to express high levels of certain proteoglycans, including DCN.